

# DistilRoBiLSTMFuse: an efficient hybrid deep learning approach for sentiment analysis

Sonia Khan Papia[1], Md Asif Khan[2], Tanvir Habib[2], Mizanur Rahman[3] and Md Nahidul Islam[4]

[1] Information Technology, Washington University of Science & Technology, Alexandria, VA, United States of America
[2] International Relations, University of Dhaka, Dhaka, Bangladesh
[3] School of Computer Science, Western Illinois University, Macomb, IL, United States of America
[4] Faculty of Electrical and Electronic Engineering, Universiti Malaysia Pahang Al-Sultan Abdullah, PEAKN, Malaysia

## ABSTRACT

In today's modern society, social media has seamlessly integrated into our daily routines, providing a platform for individuals to express their opinions and emotions openly on the internet. Within this digital domain, sentiment analysis (SA) is a vital tool to understand the emotions conveyed in written text, whether positive, negative, or neutral. However, SA faces challenges such as dealing with diverse language, uneven data, and understanding complex sentences. This study proposes an effective approach for SA. For this, we introduce a hybrid architecture named DistilRoBiLSTMFuse, designed to extract deep contextual information from complex sentences and accurately identify sentiments. In this research, we evaluate our model's performance using two popular benchmark datasets: IMDb and Twitter USAirline sentiment. The raw text data are preprocessed, and this involves several steps, including: (1) implementing a comprehensive data cleaning protocol to remove noise and unnecessary information from the raw text, (2) preparing a custom list of stopwords to retain essential words while omitting common, non-informative words, and (3) applying Lemmatization to achieve consistency in text by reducing words to their base forms, enhancing the accuracy of text analysis. To address class imbalance, this study utilized oversampling, augmenting minority class samples to match the majority, thereby ensuring uniform representation across all categories. Considering the variability in preprocessing techniques across previous studies, our research initially explores the efficacy of seven distinct machine learning (ML) models paired with two commonly employed feature transformation methods: term frequency-inverse document frequency (TF-IDF) and bag of words (BoW). This approach allows for determining which combination yields optimal performance within these ML frameworks. In our study, the DistilRoBiLSTMFuse model is evaluated on two distinct datasets and consistently delivers outstanding performance, surpassing existing state-of-the-art approaches in each case. On the IMDb dataset, our model achieves 98.91% accuracy in training, 94.16% in validation, and 93.97% in testing. The Twitter USAirline Sentiment dataset reaches 99.42% accuracy in training, 98.52% in validation, and 98.33% in testing. The experimental results clearly demonstrate the effectiveness of our hybrid DistilRoBiLSTMFuse model in SA tasks.

Corresponding author
Mizanur Rahman,
mizancse7462@gmail.com

The code for this experimental analysis is publicly available and can be accessed *via* the following DOI: https://doi.org/10.5281/zenodo.13255008.

# INTRODUCTION

The explosion of social media in recent years has transformed how we communicate, creating a vast sea of textual data from tweets, posts, and comments. This digital transformation presents both a goldmine and a labyrinth for those looking to understand the nuances of public sentiment. Sentiment analysis (SA), a key aspect of natural language processing (NLP), dives into this deep ocean of words to extract meaningful insights about how people feel and think. In the study of *Wang et al. (2022)*, sentiment analysis's impact stretches far and wide, touching everything from marketing campaigns to customer service strategies, from gauging political climates to monitoring public health trends. It is a tool that has become indispensable in our increasingly digital world.

Where SA becomes truly invaluable is in its application across various industries. For businesses, it is like having a direct line to the consumer's mind, allowing them to tailor products, services, and messages to meet the market's ever-changing demands. It provides a real-time pulse on public opinion for governments and non-profits, enabling more responsive and effective policies and communications (*Alvi et al., 2023*). Moreover, by identifying shifts in public mood or the early rumblings of social movements, SA offers a proactive means to engage with and respond to global events. In essence, SA has become a crucial navigator for navigating the complex web of digital communication, empowering decision-makers with the insights needed to act swiftly and accurately in a world where public opinion can shift at the speed of a tweet (*Chinnalagu & Durairaj, 2021*).

Sentiment analysis researchers face several challenges, including those related to the complex structure of human language and the dynamic features of communication *via* the internet. These challenges include accurately detecting and interpreting sarcasm and irony, adapting models to the nuanced expressions found across different domains, and addressing the continuous evolution of language with the emergence of new words and slang. Additionally, managing data sparsity and class imbalance and ensuring privacy and ethical considerations in data handling further complicate developing and applying SA tools. The studies conducted by *Pozzi et al. (2017)*, *Birjali, Kasri & Beni-Hssane (2021)*, *Almalki (2022)* and *Xu, Chang & Jayne (2022)* highlight, emphasize, or stress the need for domain-specific methodologies, advanced linguistic models, and ethical guidelines to tackle these challenges effectively. As the SA task continues to grow in importance across various sectors, from marketing to public health, overcoming these hurdles remains a critical focus for advancing the field and harnessing the full potential of SA in our increasingly online world.

The initial steps in SA typically involve preprocessing. Preprocessing is a critical phase where data is cleaned and normalized to enhance model performance. Techniques such as negation handling, stemming, and the removal of stop words and noise are essential for refining data quality. In *Duong & Nguyen-Thi (2021)*, they highlighted the impact of preprocessing techniques on the performance of SA models, especially in languages with limited datasets. After preprocessing, feature extraction (FE) methods are employed to extract informative information from the data. Researchers have utilized several methods for FE, such as the bag of words (BoW) model, term frequency-inverse document frequency (TF-IDF), Word2Vec, Global Vectors for Word Representation (GloVe), *etc.* (*Srivastava, Bharti & Verma, 2021*; *Das & Chakraborty, 2018*; *Lou, 2023*).

The advent of machine learning (ML) has significantly transformed SA, utilizing various algorithms to classify sentiments from text data. Among these, support vector machines (SVM), logistic regression (LR), naïve Bayes (NB), decision trees (DT), AdaBoost, and the Passive-Aggressive (PA) classifier are prominent (*Kilicoglu et al., 2019*; *Ruz, Henríquez & Mascareño, 2020*; *Ondara et al., 2022*).

The exploration of SA within the domain of social media presents a complex challenge due to the inherently dynamic and unstructured nature of online linguistic expressions. These expressions often include slang, emojis, abbreviations, and other informal linguistic constructs, which traditional NLP models struggle to interpret accurately. However, recent advancements in DL technologies have paved the way for more sophisticated models that excel in decoding these nuanced linguistic features. Among these, convolutional neural networks (CNNs) and recurrent neural networks (RNNs) stand out for their capacity to capture and analyze textual data with remarkable precision and adaptability (*Fithriasari, Jannah & Reyhana, 2020*). CNNs, initially renowned for their application in image recognition tasks, have been adeptly repurposed for text analysis in SA tasks. Their ability to detect patterns across different parts of the input data makes them particularly effective at understanding the context and significance of specific words or phrases within larger text blocks. On the other hand, RNNs, with their sequential data processing capability, excel at grasping the dynamic nature of language as it unfolds over time, making them especially suited for analyzing sentences and longer text passages where context and order significantly impact meaning.

The advent of transformer-based models, such as bidirectional encoder representations from transformers (BERT), represents a significant leap forward in SA. Contrary to their predecessors, transformers avoid sequential data processing in favour of employing attention mechanisms. These mechanisms assess the significance of various words within a sentence, irrespective of their positional order. This enables a deeper understanding of context and sentiment polarity, significantly enhancing the model's ability to interpret the subtleties of human language with higher accuracy (*Tan et al., 2022b*).

SA remains a significant challenge in NLP, primarily due to the intricate task of decoding long-distance dependencies and navigating the extensive lexical diversity found in text. Recent advancements in ML have catalyzed the development of hybrid models that combine the computational prowess of transformer models with the sequential insight provided by sequence models. Specifically, our innovative approach utilizes the DistilRoBERTa model, a

streamlined variant of the RoBERTa transformer known for its efficiency and effectiveness in generating discriminative word embeddings, which are crucial for capturing the nuanced semantics of text essential for accurate SA.

Our hybrid model amplifies its capability by integrating BiLSTM networks. These LSTMs excel at modeling the temporal dynamics of text, effectively encoding long-distance dependencies that are vital for comprehending the context and the overall sentiment of extended passages. This feature complements the transformer architecture's ability to process and understand text in parallel, enhancing the model's aptitude for identifying sentiment-bearing phrases and syntactic structures through sequential processing.

To address the challenges posed by lexical diversity and potential dataset imbalances, we employ techniques such as oversampling rather than traditional data augmentation through synonym replacement. This strategy ensures a broader representation of linguistic expressions and nuances, enriching the model's training data and bolstering its generalization capabilities. By harmonizing the transformer architecture's robust text processing with the sequential depth and temporal sensitivity of LSTMs, the model achieves a balanced and effective approach to SA. It is capable of efficiently processing large volumes of text while retaining the nuanced understanding required to assess sentiment accurately. This combination of technologies represents a comprehensive solution to the multifaceted challenges of SA, leveraging the strengths of each approach to achieve superior performance. The major contributions of this paper can be categorized into three main areas:

- In this study, we introduce a hybrid DL architecture, DistilRoBiLSTMFuse, that integrates the strengths of the DistilRoBERTa and BiLSTM for enhanced SA performance. The model employs DistilRoBERTa for its efficient contextual embedding capabilities, capturing nuanced textual features through self-attention mechanisms. This is complemented by the sequential processing strength of the BiLSTM, which adeptly manages long-range dependencies in text data, a critical aspect for understanding the full context of sentiments. Integrating DistilRoBERTa's ability to discern intricate contextual clues with the BiLSTM's proficiency in capturing temporal sequence dynamics results in a model that sets a new standard for accuracy and efficiency in SA, showcasing the potential of hybrid architectures in advancing the field.

- A comprehensive data preprocessing strategy, encompassing a thorough data cleaning protocol, custom stop words removal, and lemmatization, is applied in our SA approach. Our customized stop words list is carefully crafted to retain essential terms associated with SA, preserving each sentence's original meaning and related sentiment. Furthermore, we implement a data augmentation technique to address the challenge of an imbalanced dataset. These steps are essential for enhancing the representation of minority classes, providing a broader spectrum of data to the model, and ensuring that the input data is of the highest quality. Consequently, this careful preprocessing contributes to more accurate SA outcomes, strengthening the reliability and effectiveness of our analytical efforts.

- In this experimental study, we apply seven widely recognized ML models alongside our proposed hybrid DL model to two publicly accessible benchmark datasets. We further

compare the performance of our approaches with those from recent studies to evaluate the effectiveness and reliability of our proposed model for SA tasks. The experimental findings underscore the efficiency of our model and its adaptability and resilience across different types of textual content and SA challenges.

In 'Related Work', we thoroughly analyze the existing literature, focusing on ML and ML approaches used in SA. 'Methodology' provides a comprehensive analysis of our research methodology, including a thorough explanation of the data preprocessing techniques, feature extraction techniques, the building procedure of our proposed hybrid model, and the details of the parameter settings of the proposed model. 'Models Parameter Settings' thoroughly examines our findings, comparing them to previous research on the same datasets. Finally, this paper concludes with a section where we summarize the findings and contributions of the study, as well as the potential for future research in the field.

## RELATED WORK

Several studies have been conducted in the rapidly evolving social media SA field, highlighting significant advancements in applying FE, ML and DL techniques. Exploring these methodologies has demonstrated significant potential for accurately interpreting the substantial quantities of opinionated content produced on various social media platforms.

### Feature extraction methods

Researchers have explored various feature extraction (FE) methods for sentiment analysis, each offering distinct advantages and limitations. One prominent method is Word2Vec, introduced by *Mikolov et al. (2013)*, which represents words as continuous vectors in a high-dimensional space, capturing semantic relationships between words. This method is effective for capturing context and word similarity, making it widely applicable in various NLP tasks due to its ability to create meaningful word embeddings.

Another widely used method is GloVe, developed by *Pennington, Socher & Manning (2014)*. GloVe generates word embeddings by aggregating global word-word co-occurrence statistics from a corpus. This method effectively captures both local and global semantic relationships, making it robust for various text-mining applications. BERT, introduced by *Devlin et al. (2018)*, employs a transformer architecture to understand the context of words in a sentence bidirectionally. BERT's ability to capture nuanced meanings and relationships between words has led to state-of-the-art performance in many NLP tasks.

Despite the effectiveness of these advanced methods, more straightforward techniques such as the BoW model and TF-IDF remain popular due to their simplicity and interpretability. The BoW model converts text into a vector of word frequencies, disregarding grammar and word order but maintaining multiplicity. Each document is represented as a vector of the counts of each word's occurrences, making it particularly useful for initial exploratory studies and baseline models (*Srivastava, Bharti & Verma, 2021*). TF-IDF, on the other hand, measures the importance of a word in a document relative to a corpus, reducing the impact of frequently occurring words that are less informative. This method enhances the ability to distinguish between important and

unimportant words in the text. In *Das & Chakraborty (2018)*, they demonstrated that TF-IDF can significantly improve the accuracy of sentiment classification when combined with appropriate algorithms.

Due to their simplicity, ease of interpretation, and proven effectiveness, BoW and TF-IDF are widely used for this study. These methods provide a solid foundation for feature extraction, allowing for easy implementation and a transparent understanding of how the model processes text data. Additionally, they serve as reliable benchmarks for evaluating the performance of more complex models. Furthermore, BoW and TF-IDF are computationally efficient, making them feasible for studies with limited resources.

## Machine learning based approaches

Researchers have used different ML methods for SA task analysis. A seminal work by *Alatabi & Abbas (2020)* demonstrated the effectiveness of an SA system based on the Bayesian Rough Decision Tree (BRDT) algorithm, achieving an impressive accuracy of over 95%. This study underscored the potential of ML techniques in extracting nuanced sentiment from social media texts, marking a significant stride in the computational understanding of user-generated content.

Further investigation into the domain by *Omuya, Okeyo & Kimwele (2021)* explored the performance of naïve Bayes (NB), SVM, and K-nearest neighbor (kNN) algorithms in a SA model. Their findings suggested that ML approaches could be highly effective, with NB and SVM algorithms exhibiting high accuracy. This research added to the growing body of evidence that ML techniques could provide robust frameworks for SA capable of navigating the complex landscape of social media data.

In the study proposed by *Jagdale, Shirsat & Deshmukh (2019)* on SA on product reviews using ML techniques, they reported that naïve Bayes and support vector machine algorithms had classified product reviews with accuracies of 98.17% and 93.54%, respectively, for camera reviews. This study demonstrated the effectiveness of ML techniques in analyzing sentiment from product reviews on platforms like Amazon.

*Başarslan & Kayaalp (2021)* analyzed social media review datasets with a DL approach and achieved accuracy rates in the range of 81%–90% for the IMDb dataset and 75%–79% for the Twitter dataset using various ML and DL methods. The BERT word embedding method showed the best performance.

*Jayakody & Kumara (2021)* conducted an experiment for analyzing sentiment on product reviews on Twitter using ML approaches and found that the LR + CountVectorizer combination had achieved the highest accuracy rate of 88.26%. This underscored the potential of combining ML algorithms with text vectorization techniques for effective SA.

In another study conducted by *Kumar & Wahid (2021)* on social media analysis for sentiment classification using gradient boosting (GB) machines, XGBM outperformed other models, showcasing the effectiveness of gradient boosting machines in SA on social media datasets extracted from Kaggle.

*Abd El-Jawad, Hodhod & Omar (2018)* comprehensively compared ML and DL algorithms for SA on social media and showed a maximum accuracy rate of 83.7%,

highlighting the efficiency of hybrid learning approaches over standard supervised approaches.

*Pate et al. (2023)* implemented an SA model using ML techniques for tweets, achieving the highest accuracy of 94.65% with the SVM algorithm. This research emphasized the importance of SA in understanding public sentiment on social media platforms.

*Madhuri (2019)* explored the effectiveness of C4.5, naïve Bayes (NB), SVM, and RF on Indian Railways tweets, showcasing a diverse toolkit for handling social media text. Similarly, *Jung et al. (2016)* applied multinomial naïve Bayes (MNB) to the Sentiment140 dataset, demonstrating good performance for large-scale data. Furthermore, *Rahat, Kahir & Masum (2019)* utilized support vector classifier (SVC) and MNB for SA of airline reviews, providing insights into customer sentiments.

*Alatabi & Abbas (2020)* used the BRDT algorithm, achieving an accuracy of more than 95%. This study showcased the success of ML techniques in SA on social media.

*Hariguna & Ruangkanjanases (2023)* developed a multi-output classification framework for sentiment analysis by integrating ten distinct algorithms: BernoulliNB, decision tree, kNN, logistic regression, LinearSVC, bagging, stacking, random forest, AdaBoost, and ExtraTrees. Their objective was to assess the optimal performance and utility of each algorithm within this model. Utilizing customer review data from the cryptocurrency sector in Indonesia, the study found that LinearSVC and Stacking surpassed the remaining algorithms, each recording a notable accuracy rate of 90%. The model demonstrated an average accuracy of 88%, which was deemed satisfactory.

For the initial investigation, seven different machine learning models are employed: LR, SVM, PA, RF, AdaBoost, MNB, and XGBoost. These models are chosen based on their demonstrated effectiveness in prior research and their capability to address the complexities of sentiment analysis tasks. LR is employed for its simplicity and efficiency in binary classification problems, making it an effective baseline model. Support vector machine is included for its robustness and high accuracy in high-dimensional spaces. The PA algorithm is chosen for its suitability in online learning scenarios where data arrive sequentially. RF, known for its versatility, is used to reduce overfitting through the averaging of multiple DT. AdaBoost, an ensemble technique, is selected for its ability to improve accuracy by combining weak classifiers into a strong classifier. Multinomial naïve Bayes is included due to its effectiveness in text classification problems, given its assumption of word independence. Finally, XGBoost is chosen for its superior performance in various machine learning competitions, thanks to its powerful gradient-boosting algorithm.

Experiments with these models evaluate their performance in the context of sentiment analysis. Each model's parameters are meticulously tuned to optimize key metrics such as accuracy, precision, recall, and F1 score. The results indicate that these models provide a robust framework for sentiment analysis, each contributing unique strengths to the classification task. The inclusion of these diverse models ensures a comprehensive evaluation and comparison, leading to more reliable and generalizable findings. This approach allows for the identification of the most effective model or combination of models for the specific sentiment analysis application. Additionally, these models are used to verify the effectiveness of the dataset, preprocessing methods, and feature extraction

techniques employed in the study, ensuring that both the data preparation and the selected machine learning models adequately address the complexities of sentiment analysis.

## Deep learning based approaches

DL methods have become increasingly popular for sentiment analysis tasks. *Xiang (2021)* proposed a model that used LSTM networks, showcasing a marked improvement in prediction accuracy over traditional ML methods. This model's success illustrated DL's capability to capture the intricate sentiment patterns in social media communications, offering a more nuanced and context-aware analysis.

Similarly, a study by *Wang et al. (2022)* emphasized the advantages of DL techniques, particularly in automatic feature extraction and the rich representation of social media sentiments. These models' ability to process large datasets and their adaptability to various SA tasks underscored the transformative impact of DL on the field. The research by *Başarslan & Kayaalp (2021)* utilized Yelp restaurant reviews, IMDb movie reviews, and Twitter data, applying Word2Vec, GloVe, and bidirectional encoder representations from transformers (BERT) for word embedding. They compared the performance of CNN, long short-term memory (LSTM), recurrent neural networks (RNNs), SVM, and NB models. The accuracy ranged from 81%–90% for the IMDb dataset and 85%–98% for the Twitter dataset, demonstrating the superior performance of BERT in SA. *Lubis, Fatmi & Witarsyah (2023)* addressed the challenges of noisy data and sarcasm detection in social media SA using a DL approach based on the LSTM algorithm. Their study achieved an accuracy of 0.92, precision of 0.83, recall of 0.76, and an F1 score of 0.78 for SA, while sarcasm detection with an RNN model reached an accuracy of 0.94. This underscored the capability of LSTM in handling complex SA tasks, including sarcasm detection.

*Agüero-Torales, Salas & López-Herrera (2021)* provided an overview of DL for multilingual SA from 2017 to 2020 focused on cross-lingual and code-switching approaches. They noted a shift towards more complex architectures, like transformers, despite a stagnation in simpler models. Their findings highlighted the evolving landscape of DL in SA across various languages. *Anbukkarasi & Varadhaganapathy (2020)* employed DBLSTM for SA of self-collected Tamil tweets, leveraging the sequential nature of text data. *Alahmary, Al-Dossari & Emam (2019)* utilized BiLSTM for SA of Saudi tweets, showcasing the efficacy of RNN in capturing temporal dependencies. Additionally, *Thinh et al. (2019)* applied CNN for SA of IMDb reviews, demonstrating the capability of CNNs in capturing spatial patterns within text data. *Dashtipour et al. (2018)* developed deep autoencoders and deep CNNs for SA on a novel Persian movie reviews dataset, comparing these models with a shallow MLP. Their DL models demonstrated enhanced performance, highlighting the potential of DL for SA in languages other than English. *Singh et al. (2022)* presented a DL approach for analyzing COVID-19-related Twitter data using LSTM-RNN with attention layers for feature weighting. They significantly improved performance metrics, with an accuracy increase of 20% compared to existing approaches. The study underscored the efficiency of DL with attention mechanisms in classifying sentiments from COVID-19 tweets.

*Li et al. (2023)* presented the sentiment analysis model DC-CBLA in the travel industry combining CNN, BiLSTM, and an attention mechanism to handle complex language in customer reviews. Using BERT for accurate word vectors, it integrates local and global text features and emphasizes important data through attention. Tested on two datasets, it outperforms baselines with accuracies of 95.23% and 89.46%, as well as F1 scores of 97.05% and 93.86% for tourist attractions and hotels. In Table 1, we highlight previous studies on sentiment analysis, detailing the methods implemented and the datasets utilized in their research.

## METHODOLOGY

In this experimental analysis, we develop an approach for SA that utilizes two benchmark datasets: the IMDb movie reviews and Twitter USAirline. Our methodology encompasses a series of structured steps to enhance the model's performance in interpreting nuanced expressions of sentiment across different platforms.

Initially, we collect (online) and load the datasets, which provide a rich spectrum of sentiments from various contexts. Subsequently, we meticulously preprocess the data, employing comprehensive data cleaning protocols, custom removal of stopwords, and lemmatization to refine the quality of input for our analytical models. In the second phase, the datasets are divided, with a portion designated for training and the remainder for testing, ensuring that our model is accurate and robust when exposed to new data. Then, we prepare the labels that are critical for the model's learning process. This is followed by the pivotal task of tokenization and data preparation, converting raw text into a structured format suitable for ML algorithms. The third stage is constructing the model, where we utilize the training set to train in the model and validation checks using the testing set to ascertain the model's generalization capabilities. Finally, we check the effectiveness of the proposed model by utilizing different performance evaluation techniques. Figure 1 illustrates the flow chart of our methodology.

### Dataset
#### IMDb dataset
The IMDb dataset (*Maas et al., 2011*) is a publicly available collection of 50,000 movie reviews, curated explicitly for natural language processing and ML research, particularly in SA. This dataset is meticulously structured to ensure an even distribution between positive and negative reviews, with 25,000 instances in each category, making it an invaluable resource for developing and evaluating binary classification models. The balance between classes allows for a fair and rigorous testing ground for algorithms designed to discern positive sentiments from negative ones based on textual data. One of the key features of this dataset is its real-world applicability in understanding how sentiments can be algorithmically interpreted from textual reviews. Each review in the dataset is a consumer's genuine opinion about a movie, providing rich, nuanced language that models must navigate to classify the sentiment accurately. This challenge mirrors many real-world applications where understanding sentiment is crucial, from analyzing customer feedback

**Table 1** Overview of previous sentiment analysis studies: methods, datasets, performance, pros, and cons.

| Reference | Method | Dataset | Accuracy (%) | Pros | Cons |
|---|---|---|---|---|---|
| Madhuri (2019) | SVM | Indian Railways Tweets | 91.5 | High accuracy | May not generalize well |
| Anbukkarasi & Varadha-ganapathy (2020) | DNN | Tamil Tweets | 86.2 | Good for complex patterns | Requires large resources |
| Younas et al. (2020) | XLM-R | Roman Urdu-English Text | 71 | Handles code-mixed language | Lower accuracy |
| Jung et al. (2016) | NB | Sentiment140 | 90 | Scalable with SparkR | Naive Bayes assumptions |
| Thinh et al. (2019) | Residual CNN | IMDb | 90.02 | Effective feature extraction | Computationally intensive |
| Prabhakar et al. (2019) | Adaboost | US Airline Twitter | 85.3 | Improves accuracy | Sensitive to noise |
| Rahat, Kahir & Masum (2019) | SVM | Review Dataset | 82.48 | Robust in high dimensions | Requires tuning |
| Dhola & Saradva (2021) | BERT | Airline Reviews | 85.4 | Captures context well | High computational power |
| Wongkar & Angdresey (2019) | Naive Bayes | Twitter Data | 75.58 | Simple and fast | Assumes independence |
| Uddin, Bapery & Arif (2019) | LSTM-RNN | Bangla Social Media | 86.3 | Handles sequences well | Requires large datasets |
| Alahmary, Al-Dossari & Emam (2019) | BiLSTM | Saudi Dialect Tweets | 94 | Good for sequences | Computationally expensive |
| Dholpuria, Rana & Agrawal (2018) | CNN | IMDb Reviews | 99.33 | Excellent feature extraction | Risk of overfitting |
| Kumar & Chinnalagu (2020) | SAB-LSTM | COVID-19 Posts | 88.4 | Captures long-term dependencies | Computationally intensive |
| Saad (2020) | SVM | US Airline Twitter | 83.31 | Effective for text data | Requires tuning |

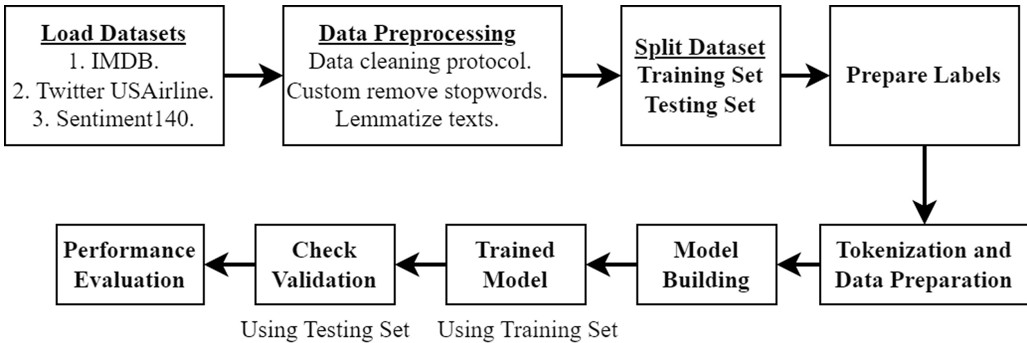

**Figure 1 Flow chart of our methodology.**

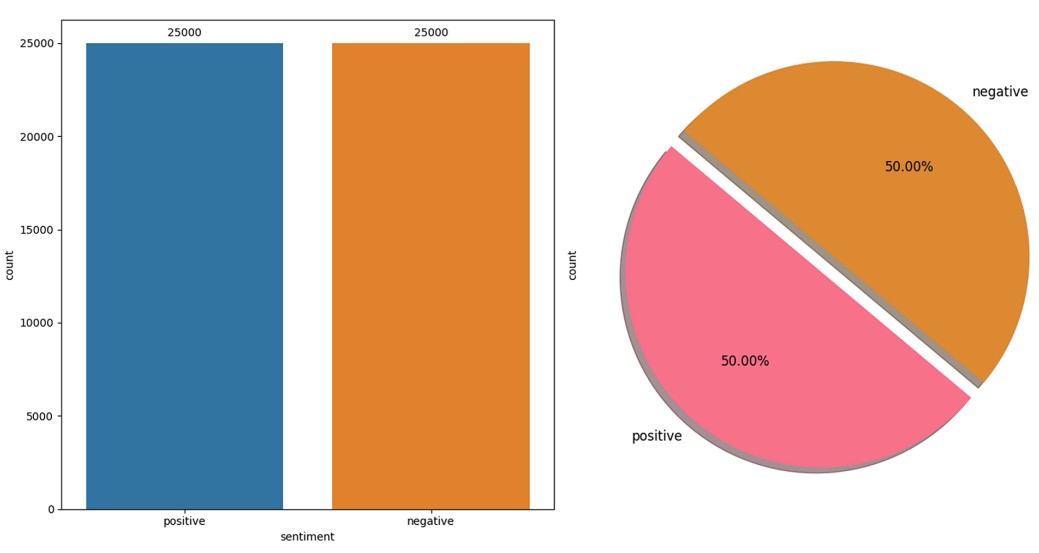

**Figure 2 IMDb dataset distribution.**

across various industries to monitoring social media sentiment in real-time. Figure 2 illustrates the data distribution of the IMDb Dataset.

### Twitter USAirline dataset

The Twitter USAirline dataset (*Crowdflower, 2015*) is a focused collection of 14,160 tweets, all related to experiences with American Airlines, captured for SA in the context of customer service and public perception. Unlike balanced datasets, this one is notably imbalanced, comprising 9,178 negative tweets, 2,363 positive tweets, and 3,099 neutral tweets. This skewed distribution reflects the real-world scenario where customers are more likely to share negative experiences on social media platforms like Twitter. The Twitter USAirline dataset serves as an invaluable resource for analyzing consumer sentiment towards airline services, offering insights into the factors that drive customer satisfaction and loyalty. It is particularly useful for developing and testing ML models to automate social media

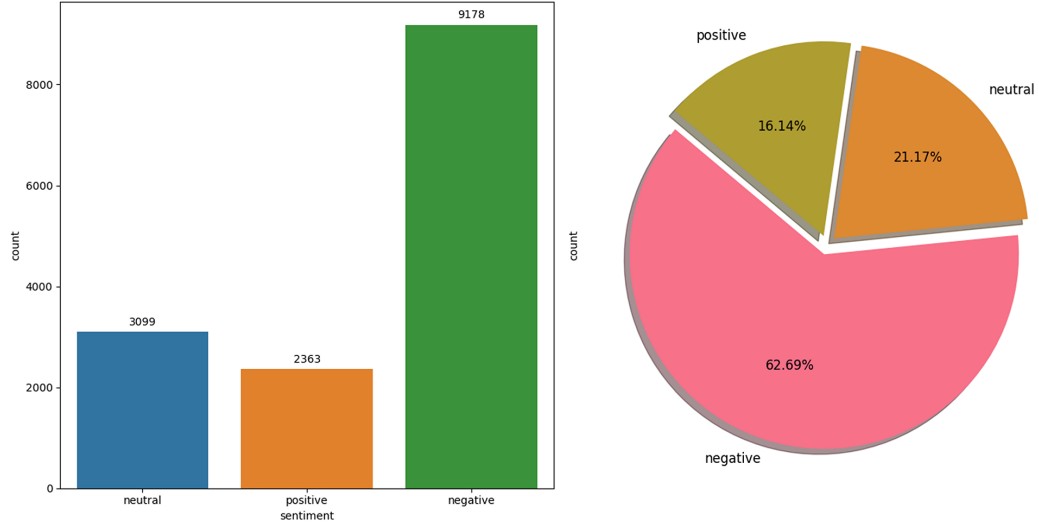

**Figure 3   USAirline dataset distribution.**

monitoring for customer feedback, which can inform service improvements and crisis management strategies. The data distribution of each class is visualized in Fig. 3.

The disparity in class distributions among the IMDb and Twitter USAirline Sentiment datasets highlights a significant challenge in SA: the impact of dataset balance on model training and performance. While the IMDb dataset naturally offers an equal representation of positive and negative sentiments, facilitating unbiased model training, the Twitter USAirline Sentiment dataset initially presents a substantial imbalance with a higher prevalence of negative tweets. This imbalance can skew model training, leading to a bias towards negative sentiment recognition, as models are more frequently exposed to negative examples.

## Dataset preprocessing
### Dataset splitting
In this study, we implement a systematic approach to data preparation to ensure robust model evaluation and prevent overfitting. For both datasets, the data is segmented into three parts: 70% for training the model, 15% for validation to ensure accuracy, and 15% for testing to measure overall performance. This splitting strategy is chosen to balance the need for a substantial amount of data for training the model while also reserving adequate data for the validation and testing phases. The training set is utilized to train the machine learning models, enabling them to acquire knowledge of the fundamental patterns in the data. The validation set is used during the model tuning phase to optimize hyperparameters and make informed decisions on model enhancements without excessively fitting to the training data. Finally, the test set is employed to analyze the final model's performance, offering an impartial evaluation of its ability to generalize to unfamiliar data.

### Data cleaning protocol

After data collection, a data cleaning protocol is applied to both datasets illustrated in Algorithm 1. In this process, the initial step involves stripping away any HTML content from the text. HTML tags are not part of the actual message or review content but are used for formatting purposes on web platforms. For instance, a review on IMDb might be submitted with bold tags to emphasize certain words, such as `great movie`, which, after HTML removal, becomes "great movie". This step ensures that the analysis focuses purely on the textual content, devoid of any web formatting distractions.

Following the cleansing of HTML, the protocol tackles emojis by converting them into descriptive text. Emojis, often used to express emotions or reactions, can carry significant weight in SA. Consider a tweet that includes a 😍 emoji to express love or like, "I love this movie 😍". The conversion process transforms it into "I love this movie heart_eyes", thereby retaining the sentiment conveyed by the emoji in a textually analyzable form.

The next step expands contractions into their full expressions, addressing texts' informal and conversational nature, especially prevalent in social media datasets. A tweet might say "I'm so excited for my trip!", which, after contraction expansion, reads "I am so excited for my trip!". This standardization is crucial for ensuring that the analysis does not misinterpret or overlook nuances in the text due to contraction use.

URLs and usernames are then removed, as they usually add little to the sentiment or content analysis and can clutter the text with irrelevant information. For example, a tweet from the *Twitter USAirline* dataset might include a mention or a link, such as "Thanks @airline for a wonderful flight! Check out the pics at www.example.com". After this cleaning step, the message is streamlined to "Thanks for a wonderful flight! Check out the pics", focusing on the sentiment and factual content.

HTML-specific line breaks are addressed next, converting them into spaces to ensure the text flows smoothly. This is particularly relevant for reviews or comments that may have used line breaks for formatting, ensuring that the transition from web format to analyzable text does not introduce unnatural breaks or gaps in the content.

Converting all text to lowercase is a critical step for standardization, ensuring that the analysis treats the same words uniformly, regardless of their original case. A sentence like "Amazing Flight Experience!" becomes "amazing flight experience!", removing any ambiguity in word recognition due to case differences.

Removing text within square brackets helps discard any annotations or extraneous information not part of the original message, often seen in IMDb reviews where users might include side notes or references. For instance, "This movie was great (citation needed)" simplifies to "This movie was great", focusing the analysis on the reviewer's opinion.

Special characters and, where necessary, digits are then stripped from the text to hone in on the words themselves. This step is vital for cleaning data from platforms like Twitter, where hashtags and special characters abound. A message like "Best movie ever!!! #excited" is cleaned to "Best movie ever excited", making it more straightforward for text analysis tools to process.

Lastly, reducing character sequences addresses the tendency in social media for users to stretch out words for emphasis, a common feature in datasets. "Sooooo excited!!!!!!" would be normalized to "Soo excited!!", ensuring that exaggerated character use does not skew the analysis.

Through these meticulously applied steps, raw text from varied datasets transforms into a clean, standardized format, ready for detailed and accurate analysis. This protocol ensures that the focus remains on the inherent content and sentiment of the texts, free from the clutter and distractions of their original formats.

### Stopwords and custom stopwords

Understanding how language conveys sentiment is crucial in the nuanced field of SA within NLP. Standard preprocessing steps often involve removing stopwords- common words that carry little to no meaningful information for analysis. However, this method can inadvertently strip away words critical for sentiment interpretation, particularly negation words like "not" and "never," which pivotally influence the sentiment conveyed. For example, removing "not" from "I am not satisfied" changes the sentiment entirely, illustrating why indiscriminate removal of stopwords can be problematic. Hence, adopting a custom approach to stopwords removal is essential for preserving the integrity of sentiment in texts.

The `custom_remove_stopwords` function represents this careful consideration by utilizing a tailored list of stopwords that spares negation and other sentiment-critical words. This approach ensures that the preprocessing enriches SA rather than detracting from it. By thoughtfully selecting which words to exclude from the stopword list, the function allows more accurate capture of sentiments expressed in various texts, such as reviews, feedback, or social media posts. This tailored method acknowledges language's complexity and context's importance, ensuring that SA remains as accurate and meaningful as possible.

### Lemmatization

Lemmatization is a crucial preprocessing step in text analysis, offering a refined approach to understanding and processing language data. Unlike the more rudimentary method of stemming, which simply trims words to a common base form often leading to inaccuracies, lemmatization considers the full morphological analysis of words. This method ensures that the derived root word, or lemma, is valid according to the language's lexicon, thus maintaining linguistic integrity. By reducing words to their lemma, lemmatization achieves a higher consistency across textual data. This is particularly beneficial for tasks that depend on accurate word frequency counts, such as identifying key themes in a corpus or conducting SA. The consistency ensured by lemmatization allows these tasks to operate on a more accurate representation of the text, thereby yielding more reliable insights.

Furthermore, lemmatization enhances the efficiency of natural language processing (NLP) tasks. By distilling words down to their base forms, it simplifies the linguistic complexity of the text, making subsequent analyses less computationally intensive and more streamlined. This process facilitates faster processing times and contributes to a deeper understanding of the text. For instance, in SA, recognizing the base form of a word

enables the algorithm to more accurately assess the sentiment being expressed, unaffected by the morphological variations of the words. Similarly, in machine translation and question-answering systems, lemmatization helps maintain the semantic correctness of the translated or generated responses, ensuring they remain true to the original text's intent. Through these applications, lemmatization is an essential process in the NLP pipeline, enhancing both the accuracy and efficiency of language-based analyses and applications.

---

**Algorithm 1** Comprehensive Data Preprocessing Steps

---

1: **function** DATACLEANINGPROTOCOL(text)
2: ⟶ **Remove HTML tags**
3: ⟶ **Convert emojis into text**
4: ⟶ **Expand contractions**
5: ⟶ **Remove URLs and usernames**
6: ⟶ **Handle HTML line breaks**
7: ⟶ **Convert text to lowercase**
8: ⟶ **Remove text within square brackets**
9: ⟶ **Remove special characters**
10: ⟶ **Reduce sequences of repetitive characters**
11: ⟶ **Remove '#' symbol from hashtags**
12: **return** cleaned text
13: **end function**

14: **function** PREPROCESSDATA(dataset)
15: **for** each text instance *text* in the dataset **do**
16: → **Apply DataCleaningProtocol**(*text*)
17: → **Apply custom_remove_stopwords**(*text*, *stopword_set*)
18: → **Apply lemmatize_texts**(*texts*, *nlp*)
19: Store preprocessed text in corresponding columns in the dataset
20: **end for**
21: **end function**

---

## Data augmentation

In our study, we focus on addressing the class imbalance within the Twitter USAirline dataset, which consists of 14,160 tweets, characterized by a significant skew towards negative sentiments—9,178 negative tweets compared to 2,363 positive and 2,619 neutral tweets. This imbalance poses a considerable challenge for training SA models to understand and classify diverse emotional expressions accurately.

To address this issue, we employ a dynamic data augmentation technique to ensure a balanced representation of sentiment classes. We begin by loading the dataset and removing any entries with missing data to maintain data consistency and quality, which are crucial for reliable ML analysis. We then assess the distribution of sentiment classes, converting the

sentiment labels into a one-hot encoded format to easily calculate the number of instances for each class.

After identifying the class with the highest number of instances, we use this information as a benchmark for balancing the other classes. For each underrepresented class, we calculate the necessary number of additional samples needed to match the maximum count found in the most populous class. Using the `resample` function from the `sklearn` library, we effectively oversample each minority class.

After augmenting the dataset, we integrate the new samples and thoroughly shuffle the data to prevent any potential order bias that could affect the training phase of our models. This rigorous approach to data preparation through resampling and shuffling lays a robust foundation for subsequent training, ultimately enhancing the performance and reliability of our SA. This method ensures that our models are trained on a dataset that reflects a more balanced view of the sentiments expressed, providing a fair representation of positive, neutral, and negative tweets.

## Features extraction
### TF-IDF

Term Frequency-Inverse Document Frequency (TF-IDF), as described by *Salton, Wong & Yang (1975)*, is a statistical method employed to determine a word's significance in a document compared to its prevalence across a broader document collection or corpus. This FE is particularly beneficial in information retrieval and text mining, including SA, where it enhances the effectiveness of algorithms in discerning and categorizing sentiments expressed in textual data. TF-IDF consists of two components: Term Frequency (TF), which calculates the frequency of a word in a document, and Inverse Document Frequency (IDF), which measures the rarity of the word across the corpus. The product of TF and IDF scores of a word gives its TF-IDF value, with higher values indicating greater importance to the document.

For instance, in the context of SA using the IMDb dataset, which consists of movie reviews, TF-IDF can distinguish between commonly used words and those that are pivotal in expressing sentiments about movies. A word like "breathtaking" in a movie review might have a high TF-IDF score, indicating a strong positive sentiment, whereas common words such as "the" would have a low TF-IDF score due to their high frequency across reviews but low discriminatory power. Similarly, by analyzing tweets from the USAirline Twitter dataset, TF-IDF can help identify terms that are uniquely expressive of customer sentiments, such as "delayed" for negative sentiments or "comfortable" for positive ones, against the backdrop of generic airline-related vocabulary.

### BoW

The Bag of Word (BoW) model is a fundamental text representation technique used in NLP and ML, especially in tasks such as SA (*Harris, 1954*; *Schütze, Manning & Raghavan, 2008*). It simplifies text content by treating it as a collection (or "bag") of individual words without considering the order or structure of those words. The technique involves creating a vocabulary of unique words from the entire dataset and then converting each

text document into a vector. Each element of this vector represents the frequency of a specific word in the document, reflecting its presence or absence in the vocabulary.

An example application of BoW can be drawn from SA on the IMDb dataset, which consists of movie reviews. Applying the BoW model transforms each review into a vector based on the occurrence of words within the review. Words that are commonly associated with positive or negative sentiments (such as "excellent" or "poor") become crucial features for training SA models. Similarly, when analyzing tweets from the USAirline Twitter dataset, the BoW model can capture frequently mentioned terms related to customer experiences, such as "delay," "cancel," or "service," which are indicative of the sentiment expressed in the tweets.

## Classification models
### SVM

Support vector machines are a type of supervised learning technique employed in classification, regression, and outlier detection tasks. Introduced in the early 1990s (*Cortes & Vapnik, 1995*), SVMs are particularly effective for complex but small to medium-sized datasets. They map data to a high-dimensional feature space to find a hyperplane that separates different class memberships, even when the data is not linearly separable. Key parameters influencing SVM performance include the kernel type (linear, polynomial, radial basis function (RBF), and sigmoid), the soft margin cost parameter (C), and the gamma parameter for non-linear classification.

SVMs have been successfully applied in various fields, including sentiment analysis. For example, SVMs were used to classify sentiments in product reviews and achieved a high accuracy of 89.98% (*Tyagi & Sharma, 2017*). Another study applied SVMs to hotel reviews and demonstrated an accuracy rate of 94% (*Simarmata & Zakariyah, 2023*).

### LR

Logistic regression (LR) is a widely used method for analyzing datasets where one or more independent variables predict a dichotomous outcome (*Kleinbaum, Kupper & Chambless, 1982*). It is widely used in fields like medicine, social sciences, and machine learning for predicting binary outcomes. LR models the probability of an outcome using the logistic function and assumes a linear relationship between the log odds of the outcome and the independent variables.

LR has been effectively used in sentiment analysis. For instance, it was applied to classify airline tweets with an accuracy of 79% (*Dhanalakshmi et al., 2023*). Another study demonstrated its application in sentiment analysis of tweets, achieving significant accuracy using logistic regression combined with FE techniques (*Jadia, 2023*).

### AdaBoost

AdaBoost, or Adaptive Boosting, is an ML meta-algorithm developed by *Freund & Schapire (1997)*. It boosts the performance of a base classifier by combining multiple weak classifiers into a strong one. Key parameters include the type of weak learners, the number of boosting rounds, and the learning rate. AdaBoost adjusts the weights of misclassified

instances, iteratively refining the model. It is popular for its simplicity and effectiveness, despite susceptibility to noise and outliers.

In *Prabhakar et al. (2019)*, they applied AdaBoost in sentiment analysis with notable success. For instance, an improved AdaBoost approach was used for analyzing US airline Twitter data, demonstrating significant performance improvements. Another study combined AdaBoost with SVM for sentiment classification on Twitter, highlighting its effectiveness in enhancing classification metrics (*Dedhia & Ramteke, 2017*).

### PA

The passive-aggressive (PA) model (*Crammer et al., 2006*) is a family of online learning algorithms designed for tasks requiring continuous updates, such as scenarios where data arrives sequentially. These algorithms make minimal updates to the model to correct mistakes (aggressive) while making no changes if the prediction is correct (passive). This approach allows the PA algorithms to adapt to new data efficiently, maintaining robustness and flexibility in dynamic environments. They are particularly effective in NLP and computer vision applications, where incremental model updates are crucial. Key parameters include the regularization parameter, which prevents overfitting by controlling model complexity, and the loss function, which quantifies prediction errors and guides updates.

Recent studies have applied the PA algorithm in various domains, demonstrating its effectiveness. For example, a study applied the PA algorithm to sentiment analysis, showing significant improvements in handling sequential data updates (*Dedhia & Ramteke, 2017*).

### XGBoost

Extreme Gradient Boosting (XGBoost) is a robust ML algorithm developed by *Chen & Guestrin (2016)*. It builds on gradient boosting, enhancing performance with parallel processing, tree pruning, and handling sparse data. Key parameters include booster type, learning rate, tree depth, and regularization terms.

In sentiment analysis, XGBoost is highly effective. *Afifah, Yulita & Sarathan (2021)* achieved 96.24% accuracy in analyzing telemedicine app reviews during the COVID-19 pandemic. *Tumuluru et al. (2023)* used an ensemble model with XGBoost for Twitter sentiment analysis, showing superior performance (*Tumuluru et al., 2023*). Additionally, *Samih, Ghadi & Fennan (2023)* and *Bhavana, Karthik & Kumari (2023)* demonstrated its effectiveness in various sentiment analysis tasks.

### MultinomialNB

MultinomialNB, or multinomial naïve Bayes, is a variant of the naïve Bayes classifier tailored for classification with discrete features, commonly used in text classification and sentiment analysis. This algorithm is effective when dealing with word frequencies and is known for its simplicity and efficiency in handling large datasets.

In sentiment analysis, MultinomialNB has demonstrated significant effectiveness. For instance, *Srivastava, Bharti & Verma (2021)* found that MultinomialNB achieved an 82% classification rate using Bag-of-Words features, outperforming other models in their study on hotel reviews (*Srivastava, Bharti & Verma, 2021*). Additionally, *Ismail, Harous &*

*Belkhouche (2016)* highlighted that MultinomialNB performed better in Twitter sentiment analysis, handling data sparsity and informal language effectively.

### RF

RF, an ensemble learning method formulated by *Breiman (2001)*, builds numerous decision trees during the training phase and delivers the class mode (for classification) or the average of predictions(for regression) from the trees. Crucial settings for this algorithm involve the tree count, the count of features to examine at each node, and the trees' maximum depth.

In sentiment analysis, Random Forest has proven effective in some task classifications. For instance, The study of *Fauzi (2018)* demonstrated its effectiveness in sentiment classification of Indonesian language texts, achieving an average OOB score of 0.829. Another study showed that Random Forest outperformed SVM in classifying product reviews from Flipkart, with an accuracy of 97% (*Karthika, Murugeswari & Manoranjithem, 2019*).

## Building procedure of proposed architecture
### DistilRoBERTa and embedding operation

DistilRoBERTa represents a significant advancement in NLP, reducing the complexities of its predecessor, RoBERTa, yet retaining its contextual embedding capabilities. The model is built on transformer blocks that employ self-attention mechanisms, enabling deep contextual understanding of token representations, which is crucial for interpreting the subtleties of language (*Sanh et al., 2019*).

### Token representation

Initially, tokens transform vector space, incorporating both lexical and positional information as follows:

$$\mathbf{E}_i = \mathbf{W}_e[\text{Token}_i] + \mathbf{W}_p[i] \tag{1}$$

where $\mathbf{E}_i$ represents the embedding of the $i$th token, $\mathbf{W}_e$ is the token embedding matrix, $\mathbf{W}_p[i]$ denotes positional encoding for position $i$, and $\text{Token}_i$ signifies the token's index within the vocabulary.

### Self-attention mechanism

The self-attention mechanism computes attention scores to ascertain the relative importance of each token within the sequence:

$$\text{Attention}(Q, K, V) = \text{softmax}\left(\frac{QK^T}{\sqrt{d_k}}\right)V \tag{2}$$

Here, $Q$, $K$, and $V$ correspond to the query, key, and value vectors, respectively, derived from the input embeddings, with $d_k$ representing the dimensionality of the keys, facilitating a dynamic aggregation of context.

### Multi-head attention

DistilRoBERTa extends the attention mechanism across multiple "heads" to diversify its focus:

$$\text{MultiHead}(Q, K, V) = \text{Concat}(\text{head}_1, \text{head}_2, \ldots, \text{head}_h)W^O \tag{3}$$

$$\text{where} \quad \text{head}_i = \text{Attention}(QW_i^Q, KW_i^K, VW_i^V) \tag{4}$$

Each head processes the input vectors through distinct sets of projection matrices $W_i^Q$, $W_i^K$, and $W_i^V$, with the outputs concatenated and further projected, enabling the model to capture complex inter-token relationships.

### *Position-wise feed-forward networks*

After the multi-head attention, each transformer block incorporates a position-wise feed-forward network:

$$\text{FFN}(x) = \max(0, xW_1 + b_1)W_2 + b_2 \tag{5}$$

This layer introduces additional non-linearity, allowing for further learning of complex patterns within the data. Through the iterative application of these mechanisms across its transformer blocks, DistilRoBERTa adeptly encodes input tokens into vectors laden with rich contextual information, facilitating advanced understanding and interpretation in downstream NLP tasks.

## Sequential processing *via* BiLSTM

After the initial embedding of tokens using DistilRoBERTa, the model employs a BiLSTM layer to enhance its understanding of the input sequence. BiLSTM networks extend the conventional LSTM architecture by processing data in both forward and backward directions, allowing the model to capture context from both past and future tokens within a sequence. This dual-direction processing is instrumental in understanding the nuanced dependencies that exist in natural language data.

### *LSTM fundamentals*

At the core of an LSTM unit are several gates that control the flow of information: the forget gate ($f_t$), input gate ($i_t$), and output gate($o_t$). These gates determine what information is retained or discarded as the sequence is processed. The LSTM cell state ($C_t$) serves as a form of memory that carries information across the sequence of tokens (*Graves & Graves, 2012*).

The operations within an LSTM cell at time step $t$ can be described by the following equations:

- **Forget gate:**

$$f_t = \sigma(W_f \cdot [h_{t-1}, x_t] + b_f) \tag{6}$$

- **Input gate:**

$$i_t = \sigma(W_i \cdot [h_{t-1}, x_t] + b_i) \tag{7}$$

$$\tilde{C}_t = \tanh(W_C \cdot [h_{t-1}, x_t] + b_C) \tag{8}$$

- **Cell state update:**

$$C_t = f_t * C_{t-1} + i_t * \tilde{C}_t \tag{9}$$

- **Output gate:**

$$o_t = \sigma(W_o \cdot [h_{t-1}, x_t] + b_o) \tag{10}$$

$$h_t = o_t * \tanh(C_t) \tag{11}$$

Here, $x_t$ is the input at the current time step, $h_{t-1}$ is the previous hidden state, $\sigma$ denotes the sigmoid activation function, and tanh is the hyperbolic tangent activation function. The weights ($W$) and biases ($b$) are parameters learned during training.

## Bidirectional processing

In a BiLSTM layer, two separate LSTM networks process the input sequence: one in the forward direction and the other in the backward direction. The forward LSTM ($\overrightarrow{LSTM}$) processes the sequence from start to end, while the backward LSTM ($\overleftarrow{LSTM}$) processes it from end to start. The outputs of both LSTMs at each time step are concatenated to form the final output for that time step:

$$h_t = [\overrightarrow{h_t}; \overleftarrow{h_t}]. \tag{12}$$

This concatenated output combines information from both past and future contexts, providing a comprehensive representation of each token within the sequence.

The application of BiLSTM layers in the model enables a richer representation of the text data, leveraging both preceding and succeeding contextual information to enhance the model's ability to understand and classify sentiment in text accurately. Through the integration of BiLSTM layers, the model acquires a dynamic capacity to capture temporal dependencies and relationships, critical for the nuanced task of SA.

### Sequential processing via BiLSTM

Following embedding, a first layer of bidirectional LSTM with 256 units captures the context in both directions of the sequence (*Schuster & Paliwal, 1997*):

$$\mathbf{H}_i = \overrightarrow{LSTM}(\mathbf{E}_i) \oplus \overleftarrow{LSTM}(\mathbf{E}_i). \tag{13}$$

Dropout and layer normalization are applied subsequently to enhance model performance and stability.

### Refinement via second BiLSTM layer

A second BiLSTM layer, with dropout (*Srivastava et al., 2014*) and global average pooling, further refines the contextual analysis:

$$\text{GlobalAveragePooling}(\mathbf{H}) \tag{14}$$

## Classification dense layer

The model's output stage comprises a dense layer with 256 neurons activated by the ReLU function, introducing non-linearity and enabling the learning of complex patterns within the condensed feature set (*Nair & Hinton, 2010*). A subsequent dropout layer (rate of 0.4)

precedes the final classification layer, which employs a softmax activation to produce a probability distribution over the target sentiment classes (*Bridle, 1990*):

$$\text{Softmax}(z_j) = \frac{e^{z_j}}{\sum_k e^{z_k}} \tag{15}$$

for each class $j$, where $z$ represents the input vector to the softmax function.

## Model compilation and training

The model is compiled with the popular and widely used Adam optimizer, a choice driven by its ability to adapt learning rates for different parameters and its efficiency in large-scale data and parameter scenarios. The learning rate is set to a conservative value of $1e - 5$, balancing the trade-off between learning speed and the risk of overshooting the minimum of the loss function. Categorical cross-entropy is the loss function suitable for multi-class classification tasks, given its effectiveness in measuring the discrepancy between the predicted probability distribution and the true distribution.

$$\text{Loss} = -\sum_{c=1}^{M} y_{o,c} \log(p_{o,c}) \tag{16}$$

where $y_{o,c}$ is a binary indicator that signifies whether the class label $c$ is the accurate classification for observation $o$, and $p_{o,c}$ is the predicted probability that observation $o$ is classified as class $c$.

Training involves two principal methods: checkpointing and early stopping. Checkpointing entails preserving the state of the model at each epoch where there is an enhancement in a monitored metric, such as validation accuracy. Early stopping, alternatively, terminates the training process when there is no enhancement in the observed metric after a designated number of epochs. This strategy prevents overfitting and ensures computational resources are utilized efficiently: monitors validation loss, halting training when no improvement is seen, facilitating model generalizability, and preventing overfitting. The model undergoes training over a predefined number of epochs or until the early stopping criterion is met, utilizing a batch size that balances the trade-off between memory utilization and the granularity of the parameter updates.

This model architecture demonstrates (in Algorithm 2) a potent approach to SA, showcasing the effective integration of transformer and RNN technologies for advanced text processing and classification.

## Performance evaluation metrics

For performance evaluation, several techniques have been widely applied to measure the model's effectiveness. Accuracy is calculated as the ratio of correct predictions (including both true positives and true negatives) to the total number of cases analyzed. It is defined by the formula where the sum of true positives and true negatives is divided by the total case count (TP, FP, TN, FN). A higher accuracy indicates a model that is more effective overall at classification. Accuracy is defined as:

$$\text{Accuracy} = \frac{TP + TN}{TP + FP + TN + FN}. \tag{17}$$

---

**Algorithm 2** Hybrid Architecture Building Procedure

---

**Require:** Preprocessed Training and Validation Datasets $(X_{\text{train}}, X_{\text{test}}, y_{\text{train}}, y_{\text{test}})$,

  1: Transformer model identifier,

  2: Hyperparameters: Sequence length, Batch size, Learning rate.

**Ensure:** A trained hybrid DL model for SA.

  3: **procedure** BUILDHYBRIDMODEL

  4:      Initialize the transformer model and tokenizer.

  5:      Define input layers for input IDs and attention masks.

  6:      Obtain embeddings from the transformer using input IDs and masks.

  7:      $LSTM_{\text{bi}} = \text{Bidirectional}(\text{LSTM}(embeddings))$

  8:      $Dropout_1 = \text{Dropout}(LSTM_{\text{bi}})$

  9:      $LayerNorm = \text{LayerNormalization}(Dropout_1)$

 10:      $LSTM_{\text{bi2}} = \text{Bidirectional}(\text{LSTM}(LayerNorm))$

 11:      $Dropout_2 = \text{Dropout}(LSTM_{\text{bi2}})$

 12:      $Pooled = \text{GlobalAveragePooling1D}(Dropout_2)$

 13:      $Dense_1 = \text{Dense}(Pooled, \text{activation} =' relu')$

 14:      $Dropout_3 = \text{Dropout}(Dense_1)$

 15:      $Output = \text{Dense}(Dropout_3, \text{activation} =' softmax')$

 16:      Compile the model with Adam optimizer and categorical crossentropy loss.

 17:      **return** compiled model

 18: **end procedure**

 19: **procedure** TRAINMODEL(model, data, epochs, callbacks)

 20:      Train the model on $X_{\text{train}}$ and $y_{\text{train}}$ with validation on $X_{\text{test}}, y_{\text{test}}$.

 21:      Use callbacks for model checkpointing and early stopping.

 22:      **return** trained model

 23: **end procedure**

---

Precision quantifies the accuracy of positive predictions, representing the ratio of true positives (TP) to the total number of predicted positives, which includes both true positives and false positives (FP). This metric is computed by dividing the true positives by the sum of true positives and false positives, reflecting the precision with which the model predicts positive classes. A higher precision value suggests greater accuracy in forecasting positive outcomes. Precision is defined as:

$$\text{Precision} = \frac{TP}{TP + FP}. \tag{18}$$

Recall assesses the percentage of actual positives that are accurately detected by the model. It is determined by dividing the true positives by the total of true positives and false negatives (FN), showcasing the model's capability to correctly recognize positive instances. A higher recall rate indicates a model's proficiency in identifying positive classes effectively.

**Table 2   Implemented ML models with used parameters.**

| Classifier | Default parameters |
|---|---|
| LR | $C = 1.0$, *solver = 'lbfgs', max_iter = 100, random_state = None* |
| SVM | *base_estimator=LinearSVC(random_state=0), cv=10* |
| PA | $C = 1.0$, *fit_intercept=True, max_iter=1000, random_state=0* |
| RF | *n_estimators=100, criterion='gini', max_depth=None, random_state=None* |
| AdaBoost | *base_estimator=None, n_estimators=50, learning_rate=1.0, random_state=None* |
| MultinomialNB | *alpha=1.0, fit_prior=True* |
| XGBoost | *use_label_encoder=False, eval_metric='logloss', random_state=0* |

Recall is defined as:

$$\text{Recall} = \frac{TP}{TP + FN} \tag{19}$$

The F1-score serves as the harmonic mean between precision and recall, effectively balancing the two. It merges these metrics into one unified measure, making it advantageous for evaluating models where a trade-off between precision and recall is necessary. The F1-score is defined as:

$$\text{F1} = 2 \cdot \frac{\text{Precision} \cdot \text{Recall}}{\text{Precision} + \text{Recall}} \tag{20}$$

## MODELS PARAMETER SETTINGS

In this section, we present the parameters utilized for each machine-learning model and the proposed hybrid DL model. Also, this section will delve into the parameter selection process for the proposed hybrid DL method.

Initially, we implemented seven ML classifiers on both datasets to check the performance of ML for the SA task. In Table 2, we present the default parameters of seven implemented ML classifiers utilized in our study. These classifiers encompass various algorithms, each with hyperparameters influencing model performance. By providing an overview of the default settings, we aim to offer insight into the initial configurations used in our experiments. These parameters are a starting point for our analysis and experimentation, allowing us to explore the behaviour and efficacy of different machine-learning models in our research context.

Table 3 presents a detailed breakdown of the hyper-parameter tuning process for our proposed hybrid DistilRoBiLSTMFuse model, designed for the SA. The selection of hyper-parameters is conducted to ensure an optimal balance between learning efficiency and model complexity. Various configurations are tested to determine the most effective combination that enhances model performance.

For the BiLSTM component of our architecture, units are tested at 64, 128, 256, and 512 to find the optimal structure for sequence learning. The finalized model employs a dual-layered approach with 256 units in the first BiLSTM layer and 128 units in the

**Table 3  The parameter selection of the proposed DistilRoBiLSTMFuse model.**

| Parameter | Parameter-tested | Optimal-value |
|---|---|---|
| BiLSTM unit | 64, 128, 256, 512 | BiLSTM(1): 256, BiLSTM(2): 128 |
| Optimizer | Adam, RMSProp, SGD | Adam |
| Learning rate | 0.0001, 0.00001 | 0.00001 |
| Dropout unit | 0.1, 0.2, 0.3, 0.4, 0.5 | Dropout(1): 0.1, Dropout(2): 0.2, Dropout(3): 0.4 |

second, providing a robust framework for capturing long-term dependencies within the text. The optimizer is a crucial element influencing the convergence speed and stability of the training process. Our experimentation with Adam, RMSProp, and SGD culminated in selecting the Adam optimizer, renowned for its adaptive learning capabilities. The learning rate is evaluated at two magnitudes, 0.0001 and 0.00001, with the latter being identified as the optimal value to fine-tune our model's weights without overshooting during updates. Lastly, we integrated dropout layers with tested rates ranging from 0.1 to 0.5 to mitigate overfitting and enhance the model's generalizability. The optimal dropout rates were established at 0.1, 0.2, and 0.4 for the respective dropout layers, balancing regularization and retaining essential features during training.

## EXPERIMENTAL ENVIRONMENT

We conduct our experimental analysis using Kaggle's computational resources, which include access to a Tesla T4-16 GB GPU and 32 GB of RAM. This setup is operated on a Windows 11 system, crucial for performing our experiments efficiently across different research projects. Kaggle's infrastructure meets our computational needs, facilitates collaboration, and enables easy sharing of our findings.

For our deep learning framework, we utilize TensorFlow and Keras, which are well-suited for building and training our DistilRoBiLSTMFuse model. Our data preprocessing and analysis are handled using pandas and scikit-learn. These tools collectively form the backbone of our experimental setup, ensuring robustness and reproducibility.

## RESULTS

In this section, we present a detailed experimental analysis of SA using two datasets. In order to demonstrate the efficacy of our method and investigate its potential, the results derived from each dataset are examined individually. The analysis for each dataset is split into two primary sections. First, we present and describe the implemented models, followed by a thorough comparison of their performance against our proposed model. Afterwards, we perform a comprehensive analysis, providing a detailed examination of recent studies and comparing different approaches that have been used on the same dataset to our own proposed model. The data preprocessing steps are applied in both datasets before using ML and DL models.

**Table 4  Performance of ML with the feature extraction on the IMDb.**

| Model | Accuracy | Precision | Recall | F1-score |
|---|---|---|---|---|
| TFIDF | | | | |
| LR | 0.8872 | 0.8873 | 0.8872 | 0.8872 |
| SVM | 0.8929 | 0.8929 | 0.8929 | 0.8929 |
| PA | 0.8795 | 0.8795 | 0.8795 | 0.8795 |
| RF | 0.8345 | 0.8345 | 0.8345 | 0.8345 |
| AdaBoost | 0.7963 | 0.7976 | 0.7965 | 0.7962 |
| MultinomialNB | 0.8607 | 0.8615 | 0.8605 | 0.8606 |
| XGBoost | 0.8554 | 0.8557 | 0.8555 | 0.8554 |
| BoW | | | | |
| LR | 0.8828 | 0.8828 | 0.8828 | 0.8828 |
| SVM | 0.8728 | 0.8728 | 0.8728 | 0.8728 |
| PA | 0.8703 | 0.8709 | 0.8704 | 0.8703 |
| RF | 0.8418 | 0.8418 | 0.8418 | 0.8418 |
| AdaBoost | 0.7947 | 0.7961 | 0.7949 | 0.7945 |
| MultinomialNB | 0.8445 | 0.8461 | 0.8443 | 0.8442 |
| XGBoost | 0.8559 | 0.8561 | 0.8560 | 0.8559 |

### Experimental analysis on IMDb

In the initial exploratory phase of our SA research on the IMDb dataset, we systematically apply seven ML models: LR, SVM, PA, RF, AdaBoost, MultinomialNB, and XGBoost. These models are then evaluated using two widely used feature extraction methods (TFIDF and Bow) to highlight the most effective combination for sentiment classification. The performance of each model is checked using four performance evaluation metrics (accuracy, precision, recall, and F1-score). The performance of seven ML models with two feature extraction methods is illustrated in Table 4.

This table shows that our top-performing setup combines SVM with TFIDF, where we observed the highest accuracy of 0.8929. This performance highlights that our implementation of SVM + TFIDF is highly effective at discriminating between different classes within the dataset. The effectiveness is likely due to SVM's capability to manage high-dimensional data and its robustness in maximizing the margin between class features. Among ML, our second best-performing model is LR, which also utilizes TFIDF (0.8872). This combination proves that LR, though slightly less efficient than SVM in this task, performs robustly when combined with TFIDF. This feature extraction technique enhances LR's ability to focus on the most impactful features, improving its accuracy in classifying the reviews. Conversely, the least effective model in our study is AdaBoost using BoW, where the scores range from 0.7945 to 0.7961 across all metrics. This lower performance could be attributed to AdaBoost's sensitivity to noisy data and outliers, which are less effectively managed by the BoW method than by TFIDF.

From Table 4, it is observed that while TFIDF generally leads to better outcomes with these models, TFIDF combined with SVM is the most effective approach in our dataset analysis. On the other hand, AdaBoost paired with BoW exhibits significant challenges,

**Table 5  Performance of the proposed DistilRoBiLSTMFuse model on the IMDb.**

| Class | Precision | Recall | F1-score |
|---|---|---|---|
| Negative | 0.9380 | 0.9415 | 0.9398 |
| Positive | 0.9414 | 0.9379 | 0.9397 |
| **Macro average** | 93.97 | 93.97 | 93.97 |
| **Weighted average** | 93.97 | 93.97 | 93.97 |
| **Accuracy (training)** | | 98.91% | |
| **Accuracy (validation)** | | 94.16% | |
| **Accuracy (testing)** | | 93.97% | |

underscoring the critical importance of selecting the appropriate model and feature extraction technique based on specific data characteristics.

Building upon the foundational groundwork laid by the initial ML models, our research moved into the more complex area of DL by developing a proposed hybrid model: the DistilRoBiLSTMFuse. This innovative model harnesses the strengths of DistilRoBERTa for understanding contextual relationships within text and the BiLSTM architecture for capturing long-term dependencies, thus aiming for a nuanced apprehension of SA in the IMDb dataset. The experimental result of our proposed model on the IMDb dataset is shown in Table 5.

The proposed approach demonstrates outstanding precision and recall in both the negative and positive classes, achieving scores of around 0.94 in both categories. This high level of performance indicates a robust ability to identify and classify both sentiments with almost equal proficiency correctly. The F1-Scores, which are harmonic means of precision and recall, also reflect this high accuracy, ensuring that both the retrieval and relevance aspects of the model's predictions are well-balanced. Further examining the overall accuracy, the model achieves an outstanding 98.91% in training, which shows that it effectively learns from the dataset with minimal overfitting. This high training accuracy translates well into the validation and testing phases, where it maintains a high accuracy of 94.16% and 93.97%, respectively.

When comparing our model to traditional ML models discussed earlier, such as SVM and LR using TFIDF and BoW, the DistilRoBiLSTMFuse model distinctly outperforms them in terms of both training and testing accuracy. While the best SVM model with TFIDF achieved an accuracy close to 0.8929, the DistilRoBiLSTMFuse model surpasses this, reaching 93.97% in testing accuracy. This significant improvement highlights the effectiveness of integrating advanced neural network architectures and hybrid models in handling complex tasks like SA on the IMDb dataset, making it a superior choice for achieving high accuracy and reliability in real-world applications.

Figure 4 depicts the accuracy and demonstrates the model's learning progress, which consistently improves during training and reaches a peak of 98.91% by the sixth epoch. One significant characteristic of this figure is the significant increase in validation accuracy during the third epoch, reaching a peak of 94.16%. Following this peak, the curve levels off. These findings indicate that the third epoch is when the model attains optimal equilibrium between the processes of learning and generalization. The second image illustrates the
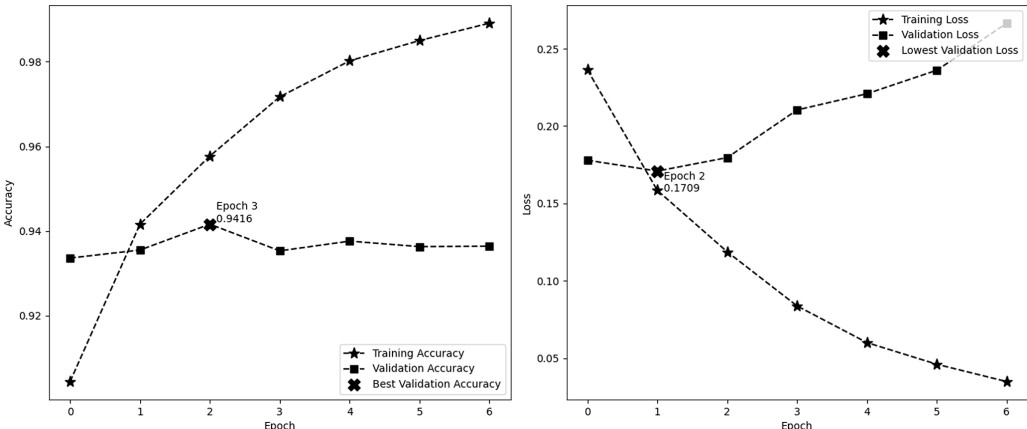

**Figure 4** Performance (accuracy and loss) curve of the proposed model on IMDb.

model's loss over epochs, showing a consistent decrease in the training loss. This decrease indicates the model's increasing accuracy in making predictions. On the other hand, the validation loss reaches its lowest point at epoch 2, with a value of 0.1709, and subsequently increases, suggesting that the model might be fitting too closely to the training data. These visual cues emphasize the significance of monitoring both accuracy and loss in order to avoid overtraining and sustain the model's performance on new data. In this experiment, we employ EarlyStopping strategies with a patience of 4. These techniques help to reduce overfitting and ensure that the model generalizes effectively to new data, all while maintaining computing efficiency.

The confusion matrix in Fig. 5 provides a comprehensive overview of the performance of our proposed model on the IMDb dataset. In a confusion matrix, the diagonal elements (top-left and bottom-right) represent the counts of true positive (3,527) and true negative (3,521) predictions, indicating instances where the model correctly identified the positive and negative classes, respectively. The high values along the diagonal demonstrate the model's effectiveness in making accurate predictions. On the other hand, the off-diagonal elements (top-right and bottom-left) correspond to the counts of false positive (219) and false negative (233) predictions. False positives arise when the model mistakenly identifies an observation as belonging to the positive class, whereas false negatives happen when the model wrongly labels an observation as negative. These off-diagonal values highlight the instances where the model's predictions were incorrect. From the confusion matrix of Fig. 5, our proposed model shows higher performance, as evidenced by the high number of correct predictions. The model has a robust capacity to classify occurrences in the IMDb dataset effectively.

Figure 6 presents our proposed model's receiver operating characteristic (ROC) curves on the IMDb dataset. The ROC curve represents the true positive rate (sensitivity) *versus* the false positive rate (1-specificity) across different threshold values. The ROC curve for the positive class is shown in orange, and the ROC curve for the negative class is shown in blue. Both curves have an area under the curve (AUC) of 0.98, indicating the model's

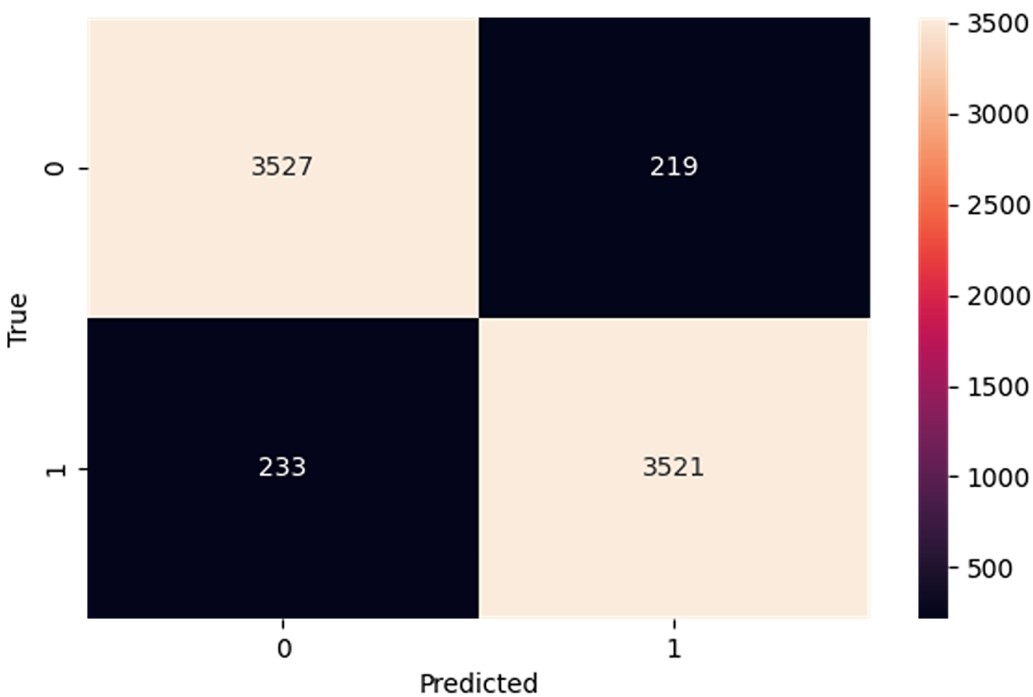

**Figure 5** **Confusion matrix on IMDb.**

excellent discriminative ability. An AUC value of 0.98 suggests that the model effectively distinguishes between the positive and negative classes. The curves near the top-left corner of the plot suggest a high level of performance, marked by a minimal occurrence of false positives and false negatives.

### Comparative analysis on IMDb

The comparative analysis provided in Table 6 presents details of various models' performances on the IMDb dataset, including ML and DL methods applied in previous studies. Traditional ML techniques such as decision trees (*Zharmagambetov & Pak, 2015*), AdaBoost (*Vadivukarassi, Puviarasan & Aruna, 2018*), logistic regression (*Dholpuria, Rana & Agrawal, 2018*), KNN (*Dholpuria, Rana & Agrawal, 2018*), and NB (*Jung et al., 2016*) have shown a range of performances, with LR and NB achieving high accuracies around 87%. On the DL side, models like GRU (*Hossen et al., 2021*), LSTM (*Hossen et al., 2021*), BiLSTM (*Garg & Kaliyar, 2020*), and combinations like CNN-LSTM (*Jain, Saravanan & Pamula, 2021*) and CNN-BiLSTM (*Rhanoui et al., 2019*) performed well but did not surpass 90% in accuracy.

On the other hand, the more advanced DL models incorporating transformer architectures, specifically RoBERTa-LSTM (*Tan et al., 2022a*) and our proposed model DistilRoBiLSTMFuse, demonstrate superior performance. The RoBERTa-LSTM model achieved an impressive accuracy of 92.96%, making it the second-best in this analysis. Our model, DistilRoBiLSTMFuse, outperformed all with an accuracy of 93.97%. This represents

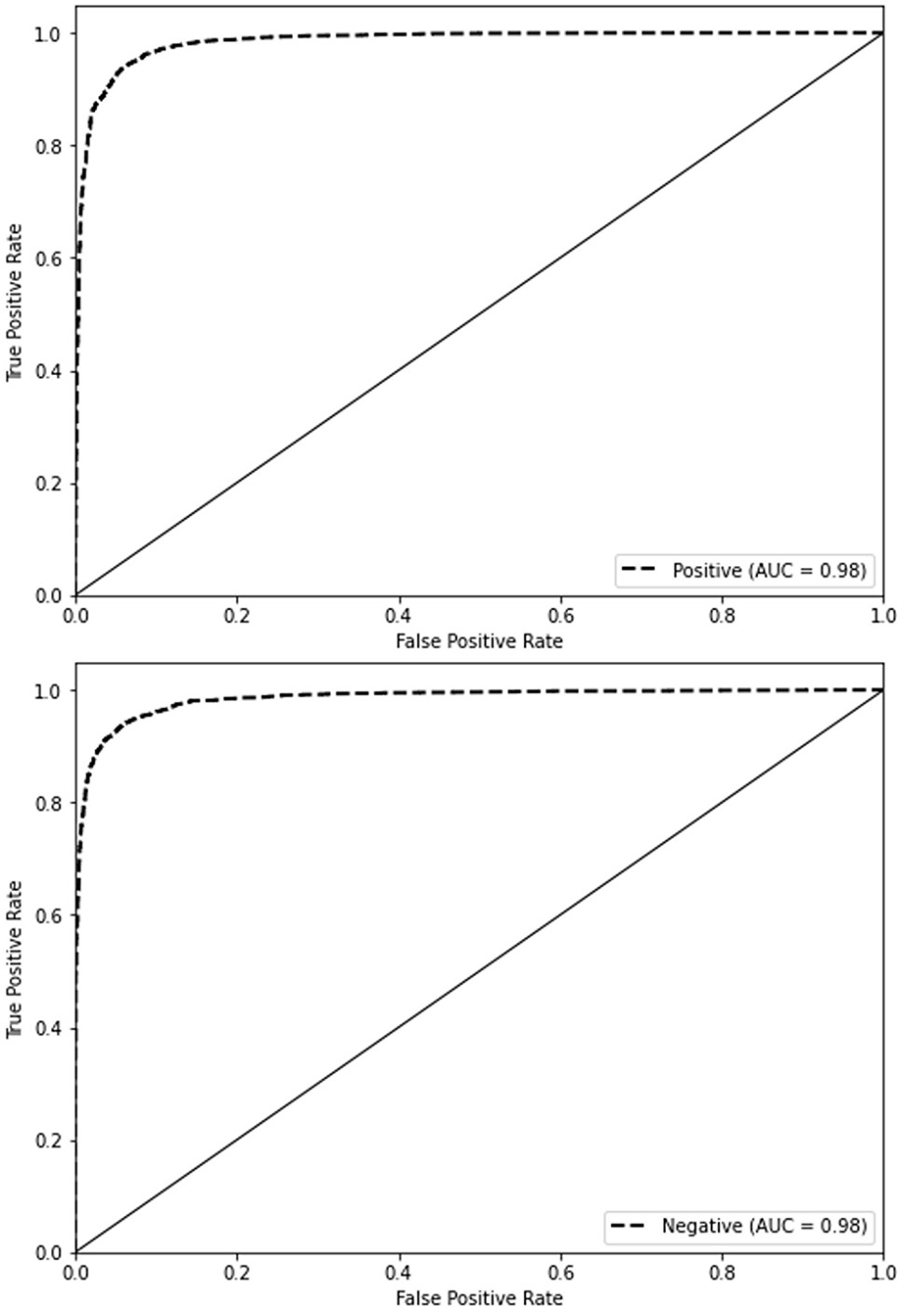

**Figure 6   ROC curve on IMDb.**

**Table 6  The comparative analysis on IMDb.**

| Methods | Accuracy | Precision | Recall | F1-score | Reference |
|---------|----------|-----------|--------|----------|-----------|
| DT | 73.46 | 74 | 73 | 73 | *Zharmagambetov & Pak (2015)* |
| AdaBoost | 83.37 | 83 | 83 | 83 | *Vadivukarassi, Puviarasan & Aruna (2018)* |
| LR | 87.12 | 90 | 90 | 90 | *Dholpuria, Rana & Agrawal (2018)* |
| KNN | 77.37 | 78 | 77 | 77 | *Dholpuria, Rana & Agrawal (2018)* |
| NB | 87.01 | 87 | 87 | 87 | *Jung et al. (2016)* |
| GRU | 87.88 | 88 | 88 | 88 | *Hossen et al. (2021)* |
| LSTM | 85.11 | 85 | 85 | 85 | *Hossen et al. (2021)* |
| BiLSTM | 86.28 | 87 | 86 | 86 | *Garg & Kaliyar (2020)* |
| CNN-LSTM | 86.07 | 86 | 86 | 86 | *Jain, Saravanan & Pamula (2021)* |
| CNN-BiLSTM | 86.16 | 86 | 86 | 86 | *Rhanoui et al. (2019)* |
| RoBERTa-LSTM | 92.96 | 93 | 93 | 93 | *Tan et al. (2022a)* |
| DistilRoBiLSTMFuse | 93.97 | 93.97 | 93.97 | 93.75 | Our proposed |

a 1.01% improvement over the RoBERTa-LSTM, indicating a significant advancement given the high-performance levels already achieved by the top models.

The superior performance of our model, DistilRoBiLSTMFuse, can be attributed to its innovative architecture that combines the distilled strengths of the RoBERTa with the sequential data processing capabilities of BiLSTM. This hybrid approach leverages the deep contextual understanding of transformer models with the proven effectiveness of bidirectional LSTM for sequence learning. The integration enhances the model's ability to discern nuanced sentiment expressions within the IMDb reviews, leading to a notable accuracy improvement of 1.01% over the second-best model, RoBERTa-LSTM. This advancement underscores the potential of combining distilled transformer models with traditional recurrent neural networks to push the boundaries of SA accuracy.

### Experimental analysis on USAirline

We also implemented the same preprocessing and ML models with two distinct feature extraction methods on the Twitter USAirline dataset (presented in Table 7). The performance mentioned in this table is achieved after using the data balancing method. Among the models implemented in this analysis, the PA Classifier + TFIDF method provides the highest performance, with scores of 0.9693 for accuracy, 0.9696 for precision, 0.9694 for recall, and 0.9695 for the F1 score. This model is adept at handling large-scale streaming data and adjusting its parameters dynamically in response to each instance's error, making it highly effective for text classification tasks. Closely following the PA classifier, the RF + TFIDF exhibited superior performance, with an accuracy of 0.9635 and an F1-score of 0.9636. Its ability to reduce overfitting by averaging multiple deep decision trees contributes to its robust performance across different datasets. On the other end of the spectrum, the MNB model demonstrated the lowest performance metrics under

**Table 7  Performance evaluation of ML models using feature extraction methods on USAirline.**

| Model | Accuracy | Precision | Recall | F1-score |
|---|---|---|---|---|
| **TFIDF** | | | | |
| LR | 0.9390 | 0.9391 | 0.9393 | 0.9392 |
| SVM | 0.9628 | 0.9629 | 0.9630 | 0.9629 |
| PA | 0.9693 | 0.9696 | 0.9694 | 0.9695 |
| RF | 0.9635 | 0.9635 | 0.9638 | 0.9636 |
| AdaBoost | 0.8471 | 0.8538 | 0.8482 | 0.8474 |
| MultinomialNB | 0.8798 | 0.8815 | 0.8803 | 0.8787 |
| XGBoost | 0.9475 | 0.9480 | 0.9479 | 0.9478 |
| **BoW** | | | | |
| LR | 0.8529 | 0.8572 | 0.8541 | 0.8532 |
| SVM | 0.8489 | 0.8527 | 0.8502 | 0.8492 |
| PA | 0.8306 | 0.8334 | 0.8318 | 0.8313 |
| RF | 0.8624 | 0.8622 | 0.8635 | 0.8615 |
| AdaBoost | 0.7990 | 0.8331 | 0.8008 | 0.7913 |
| MultinomialNB | 0.7394 | 0.7591 | 0.7412 | 0.7312 |
| XGBoost | 0.9432 | 0.9434 | 0.9436 | 0.9435 |

the COUNT transformation, with an accuracy of 0.7394 and an F1-score of 0.7312. This outcome suggests that while naïve Bayes is a simple and fast classifier, its performance may lag when the independence assumption between features is often violated, as in text data.

Since each class of this dataset is not balanced, we evaluate our proposed DistilRoBiLSTMFuse model under two distinct conditions: with and without the implementation of data augmentation techniques. This comprehensive analysis aimed to determine the impact of data augmentation on enhancing model performance. Table 8 demonstrates the performance of our proposed DistilRoBiLSTMFuse model on the USAirline Tweet dataset without data augmentation. Precision within this context measures the accuracy of the model's predictions for each sentiment class, denoting the proportion of correctly identified positive instances relative to all positive predictions made by the model. Recall captures the model's capacity to detect all pertinent instances within a specific class across the dataset. The F1-score, as a harmonic mean of precision and recall, serves as an aggregate measure of the model's accuracy, balancing both the precision and recall metrics. Table 8 indicates that the negative class achieves remarkably high precision and recall, resulting in a nearly perfect F1-score of 99.97%.

The neutral and positive classes exhibit slightly lower performance metrics but still maintain commendable scores above 85% across all three metrics, indicative of robust model performance in these categories. Furthermore, the table presents overarching performance indicators such as the macro average and weighted average of precision, recall, and F1-score, alongside a consolidated measure of Accuracy during the testing phase. The macro average stands at 91.89%, reflecting the average performance across all classes, unweighted by class frequency. The weighted average, which considers the frequency of each class, mirrors the testing accuracy, both marked at 95.50%. This high accuracy

**Table 8  Performance of the DistilRoBiLSTMFuse without augmentation on USAirline.**

| Class | Precision | Recall | F1-score |
|---|---|---|---|
| Negative | 99.95 | 100.00 | 99.97 |
| Neutral | 88.56 | 89.60 | 89.07 |
| Positive | 87.34 | 85.92 | 86.63 |
| **Macro average** | 91.95 | 91.84 | 91.89 |
| **Weighted average** | 95.49 | 95.50 | 95.50 |
| **Accuracy (testing)** | | 95.50% | |

**Table 9  Performance of the proposed DistilRoBiLSTMFuse with augmentation on USAirline.**

| Class | Precision | Recall | F1-score |
|---|---|---|---|
| Negative | 1.0000 | 1.0000 | 1.0000 |
| Neutral | 0.9826 | 0.9657 | 0.9741 |
| Positive | 0.9670 | 0.9832 | 0.9751 |
| **Macro average** | 0.9832 | 0.9830 | 0.9831 |
| **Weighted average** | 0.9834 | 0.9833 | 0.9833 |
| **Accuracy (training)** | | 99.42% | |
| **Accuracy (validation)** | | 98.52% | |
| **Accuracy (testing)** | | 98.33% | |

indicates that the model performs exceptionally well across varying class distributions, highlighting its effectiveness in handling unbalanced data typical of real-world scenarios.

After an initial analysis of our proposed model without augmentation, we conducted further analysis using data augmentation and balancing each class. Table 9 illustrates the performance of our DistilRoBiLSTMFuse model with each class balanced. After balancing, the model achieved perfect scores across all metrics for the negative class, indicating that it was both highly precise and comprehensive in identifying negative sentiments. There were no instances where the model misclassified a negative sentiment or failed to detect it. In the case of the neutral class, the scores are slightly lower but still very high, with a precision of 98.26% and a recall of 96.57%. This suggests that while the model is mostly accurate at identifying neutral sentiments, there is a small margin of error where it might confuse them with other sentiments or miss them altogether. The positive class similarly shows excellent results, with a precision of 96.70% and a recall of 98.32%. This indicates a strong ability to identify positive sentiments, though there is a slight tendency to over-identify other sentiments as positive.

The model consistently achieves precision, recall, and F1-scores near 98.3% on both macro and weighted scales, showcasing its high accuracy and adeptness at equitably managing these metrics across various classes. Accuracy metrics further reinforce the model's robustness, showing high consistency in performance across training, validation, and testing phases—99.42%, 98.52%, and 98.33%, respectively. This consistency is crucial for validating the model's effectiveness in real-world applications, demonstrating that it maintains high accuracy even on unseen data.

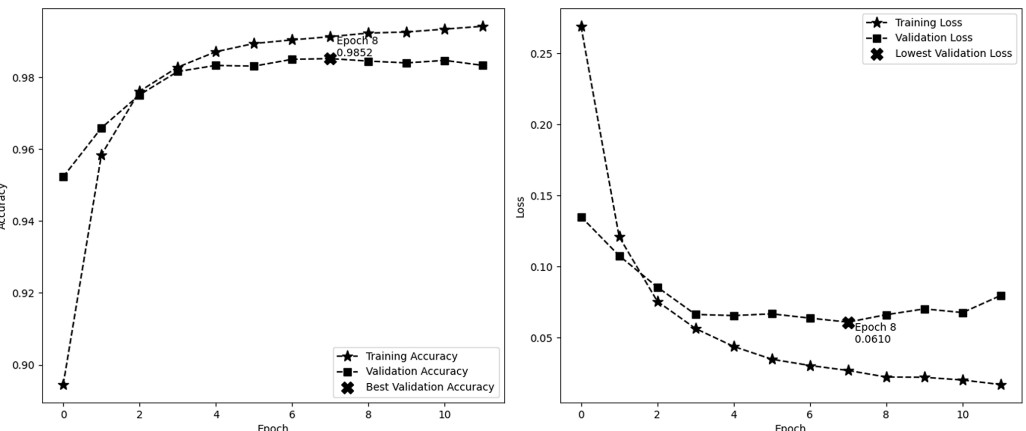

**Figure 7** Performance (accuracy and loss) curve of the proposed model on USAirline.

Figure 7 illustrates the training trajectory of an ML model, with the progression of epochs revealing a consistent increase in accuracy and a corresponding decrease in loss for both the training and validation sets. The validation accuracy ascends to a peak of approximately 98.52% around the eighth epoch, as highlighted on the graph. This peak represents the model's highest generalization performance. In parallel, the loss graph exhibits a downward slope, with both the training and validation losses diminishing as the epochs progress. The validation loss is notably minimized to its lowest at the same eighth epoch, with a value of around 0.0610. This confluence of the highest validation accuracy and the lowest validation loss at the same epoch suggests that the model has achieved an optimal balance of learning and generalizing, indicating its readiness for effective real-world applications without the complication of overfitting.

Figure 8 illustrates the confusion matrix for our model's performance on the USAirline dataset. The diagonal elements represent the counts of correct predictions, with 1,416 true positives for Class 0, 1,296 true positives for Class 1, and 1,350 true positives for Class 2, indicating the instances where the model correctly identified each class. Conversely, the off-diagonal elements represent the misclassifications. Specifically, there are 46 instances where Class 1 was misclassified as Class 2, and 23 instances where Class 2 was misclassified as Class 1. These misclassifications are relatively few in number.

Figure 9 presents the ROC curves for our model on the USAirline dataset. The ROC curves for Class 0, Class 1, and Class 2 are depicted in blue, orange, and green, respectively. Each curve has an area under the curve (AUC) of 1.00, indicating the perfect discriminative ability of the model for all classes. The curves' proximity to the top-left corner of the plot demonstrates the model's high performance in distinguishing between the different classes. The AUC values of 1.00 confirm that the model has achieved perfect classification performance, effectively separating each class without any errors.

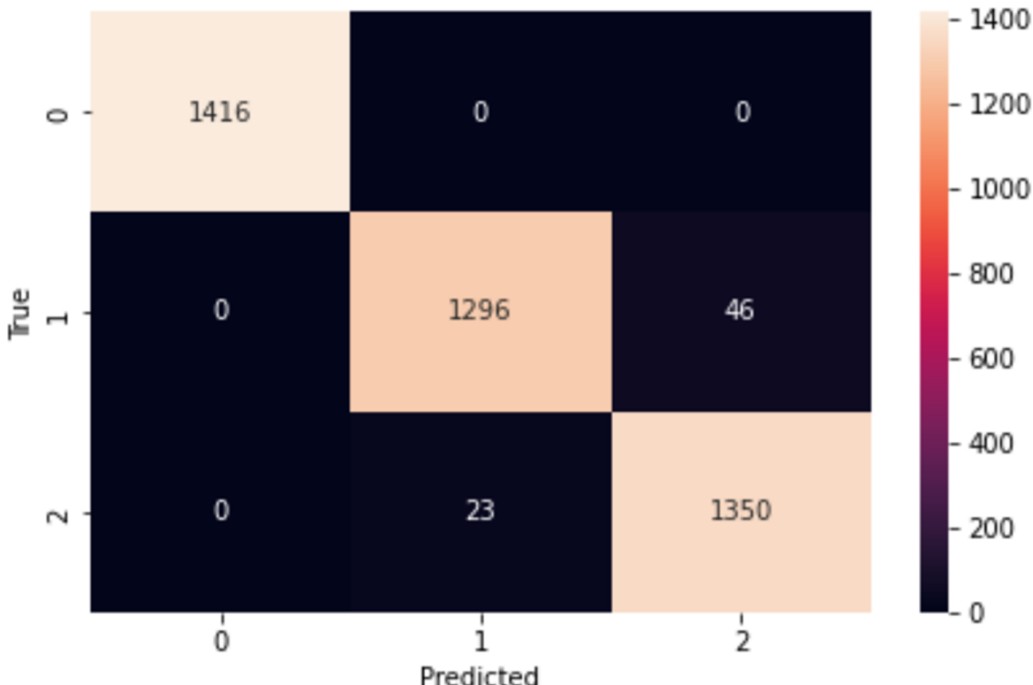

**Figure 8** Confusion matrix on USAirline.

### Comparative analysis on USAirline

Table 10 provides a comparative analysis of various methods applied to previous studies to the Twitter USAirline Sentiment dataset.

In the ML category, methods like DT, AdaBoost, LR, KNN, and NB show a range of performance, with LR outperforming others with an accuracy of 80.5% and F1-score of 72%, as reported in the cited study. Moving to the DL methods, models such as GRU, LSTM, BiLSTM, CNN-LSTM, and CNN-BiLSTM are listed, with GRU leading this group with an accuracy of 78.55% and an F1-score of 72%. The RoBERTa-LSTM model emerges as the second-best performer with significant scores across the board—91.37% accuracy, 91% precision, and a 91% F1-score—indicating a substantial improvement over the traditional DL models.

On the other hand, the proposed DistilRoBiLSTMFuse model is established as the most cutting-edge approach for this task by achieving the highest testing accuracy of 98.33%, precision of 97.34%, and an F1 score of 98.33%. It demonstrates a notable improvement over the RoBERTa-LSTM model, which is the second-best in this comparison. The gains are significant, with a 6.96 percentage point increase in accuracy and similar improvements in other metrics. Our DistilRoBiLSTMFuse model stands out for its efficiency and advanced performance, establishing a new benchmark for SA on the Twitter USAirline dataset. It reflects a well-tuned model that has successfully captured the nuances of natural language, likely due to an effective combination of distillation techniques and a robust fusion of bidirectional LSTM with a transformer architecture. The performance leap suggests that

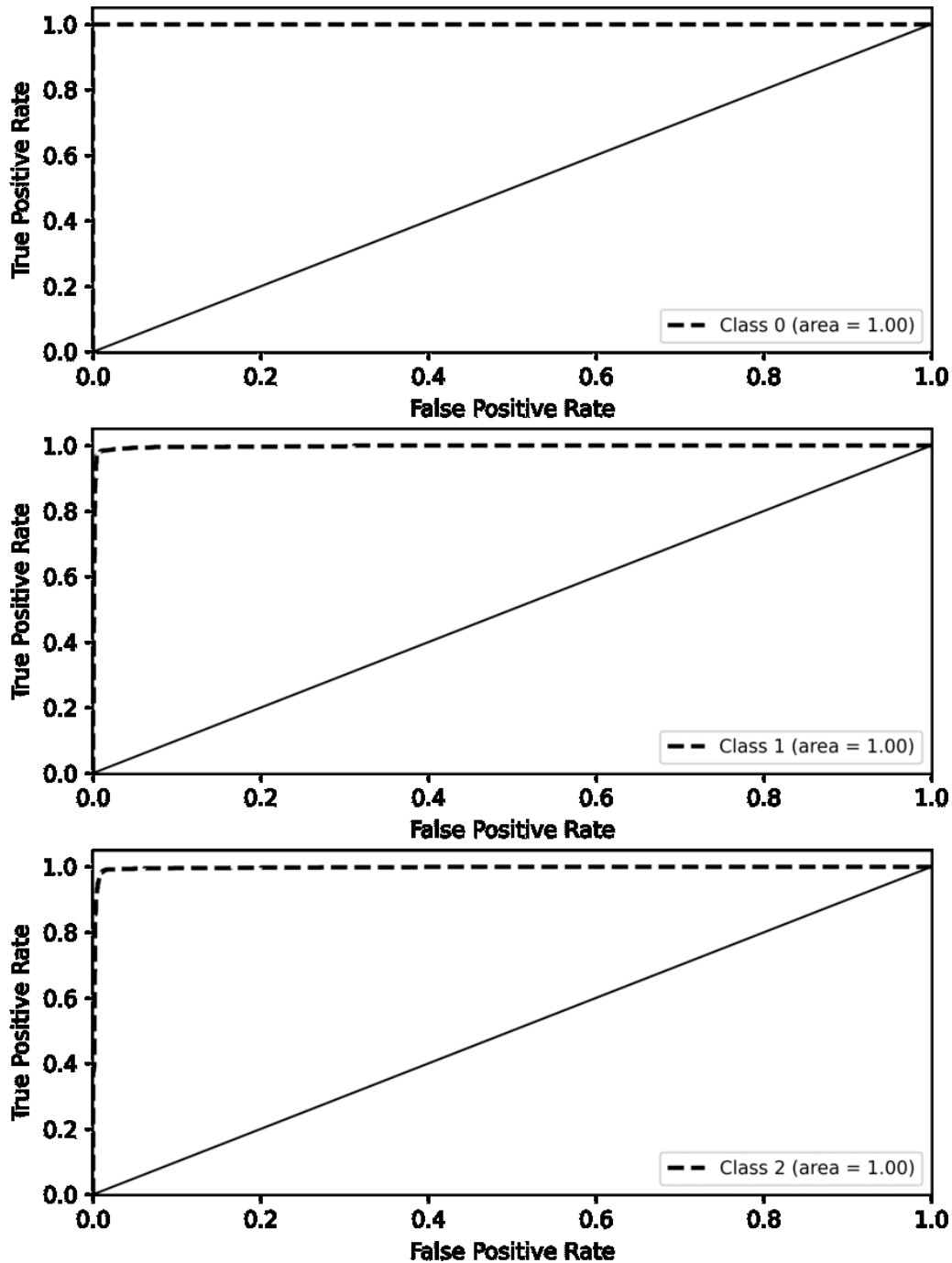

**Figure 9** ROC curve on USAirline.

**Table 10 The comparative results on the Twitter USAirline.**

| Methods | Accuracy | Precision | Recall | F1-score | Ref |
|---|---|---|---|---|---|
| DT | 71.4 | 62 | 56 | 58 | *Zharmagambetov & Pak (2015)* |
| AdaBoost | 74.59 | 67 | 63 | 65 | *Vadivukarassi, Puviarasan & Aruna (2018)* |
| LR | 80.5 | 78 | 69 | 72 | *Dholpuria, Rana & Agrawal (2018)* |
| KNN | 68.41 | 60 | 60 | 60 | *Dholpuria, Rana & Agrawal (2018)* |
| NB | 69.5 | 79 | 44 | 45 | *Jung et al. (2016)* |
| GRU | 78.55 | 73 | 71 | 72 | *Hossen et al. (2021)* |
| LSTM | 77.56 | 71 | 69 | 69 | *Hossen et al. (2021)* |
| BiLSTM | 77.46 | 71 | 69 | 70 | *Garg & Kaliyar (2020)* |
| CNN-LSTM | 76.02 | 68 | 69 | 69 | *Jain, Saravanan & Pamula (2021)* |
| CNN-BiLSTM | 77.32 | 70 | 65 | 67 | *Rhanoui et al. (2019)* |
| RoBERTa-LSTM | 91.37 | 91 | 91 | 91 | *Tan et al. (2022a)* |
| DistilRoBiLSTMFuse | 98.33 | 98.34 | 98.33 | 98.33 | Our proposed |

the model's architecture and training regime are highly effective for the specific challenges posed by SA in social media text data.

# DISCUSSION AND FUTURE WORK

## Scalability and applicability of the data preprocessing strategy
### Scalability to other datasets
The data preprocessing strategy employed in this research is designed to be versatile and adaptable, ensuring it can be readily applied to a wide range of datasets beyond the two datasets used in this experimental analysis for sentiment analysis.

This study uses several preprocessing steps to ensure the quality of the text and remove redundant information which helps the classification model to achieve optimal performance. Most of the preprocessing steps, for example: HTML Removal, Contraction Expansion, and Lowercasing, are widely applicable to most of the sentiment analysis datasets (which datasets are available online). For the Stopword Removal technique, we have created a custom comprehensive list by analyzing the datasets used in sentiment analysis. Besides, to handle contractions commonly found in informal text, we utilized a contractions expansion dictionary. This dictionary maps contractions to their expanded forms, ensuring that all textual data is standardized. The use of such a dictionary is crucial in dealing with the informal and often abbreviated nature of user-generated content (For example: "I'm" is expanded to "I am"). Our contractions expansion dictionary also includes domain-specific terms relevant to the datasets we are working with, such as abbreviations and slang commonly used in movie reviews and social media posts (for instance: "CGI" expanded to "Computer-Generated Imagery"). This approach ensures that all forms of

contractions and abbreviations are consistently developed, reducing variability in the text data and improving the effectiveness of subsequent text analysis techniques.

Some of the datasets need specific preprocessing based on their data format. For example, the Twitter US Airline dataset was processed by combining the 'negativereason' and 'text' columns into a new column named 'review,' filling any missing values in 'negativereason' with an empty string. Additionally, the 'sentiment' column was derived from 'airline_sentiment,' and the dataset was refined to include only the 'review' and 'sentiment' columns for further analysis. This tailored preprocessing step ensures that the dataset's unique structure and relevant information are adequately prepared for subsequent analysis.

To address the issue of class imbalance in our datasets, researchers commonly employ various techniques, such as the Synthetic Minority Over-sampling Technique (SMOTE). Initially, we utilized a widely implemented method, SMOTE, to assess the performance of our machine learning model. SMOTE works by creating synthetic samples for the minority classes, thereby balancing the dataset and enhancing the model's ability to learn from underrepresented classes.

However, in our proposed approach, we implemented a custom oversampling method designed to meet our specific needs. This method involves several key steps. First, we calculated the number of instances for each class and identified the maximum class count, which served as the target count for oversampling. Next, for each minority class, we determined the number of additional samples needed to match the maximum class count. We then randomly sampled instances from these minority classes with replacements to achieve the desired count. After oversampling, the dataset was shuffled to ensure that the samples were randomly distributed, a crucial step to prevent any order bias that might affect the training process. This custom oversampling technique ensures that each class in the dataset is equally represented, thus mitigating the risk of model bias towards the majority class. By balancing the class distribution, our approach enhances the model's ability to learn and generalize across all classes, improving performance and robustness.

Our approach offers several advantages over traditional methods, like SMOTE. Specifically, it avoids the potential for overfitting to synthetic samples generated by SMOTE, as it only uses real instances from the dataset. Additionally, this method is straightforward to implement and computationally efficient, making it suitable for large-scale datasets. These steps are not specific to the datasets used in this study but are fundamental preprocessing techniques applicable to various text datasets. The general nature of these steps ensures that they can be easily adapted and applied to other datasets with minimal modifications. This flexibility demonstrates the robustness and versatility of the preprocessing strategy, making it suitable for a wide array of text analysis tasks across different domains and contexts.

## Future work

Future work will focus on validating the proposed DistilRoBiLSTMFuse model with a broader range of datasets, including those from different domains and languages. This validation will help determine the model's robustness and adaptability across diverse textual data. Additionally, we aim to extend the preprocessing strategy to incorporate

advanced techniques to achieve more robust performance and applicability for more diverse datasets. Exploring the integration of domain-specific ontologies and knowledge graphs will enhance the contextual understanding of the text, especially in specialized fields.

Another important direction for future research is the development of adaptive preprocessing methods that dynamically adjust based on the dataset's characteristics, ensuring optimal preprocessing for various types of text data. Furthermore, we will explore the scalability of our approach to multilingual datasets, addressing the challenges of processing text in different languages and scripts to create more inclusive and versatile sentiment analysis models.

Finally, incorporating unsupervised and semi-supervised learning techniques will be investigated to improve the model's performance on datasets with limited labelled data, enhancing its applicability in real-world scenarios. By exploring these advanced methods, we aim to develop a more robust and versatile sentiment analysis framework that can adapt to various datasets and provide high accuracy and reliability in different applications.

## CONCLUSION

This study has successfully demonstrated the efficacy of the proposed hybrid architecture, DistilRoBiLSTMFuse, for SA. By utilizing the potent combination of distilled transformers and BiLSTM, our model handles the complexities of language in social media texts. The model underwent meticulous evaluation using the IMDb and Twitter USAirline sentiment benchmark datasets, including rigorous preprocessing and data augmentation to ensure robustness against class imbalances and to refine its interpretive abilities. Our comparative analysis shows that our proposed model, DistilRoBiLSTMFuse, outperforms seven traditional ML models and two feature transformation methods. It achieves an accuracy of 94.16% on the validation and 93.97% on the testing set using the IMDb dataset. Impressively, it also records 98.52% validation and 98.33% testing accuracy on the Twitter USAirline Sentiment dataset. This performance establishes a new standard for state-of-the-art SA methods. These results underline our model's potential as a transformative tool for understanding and analyzing online sentiment, providing significant implications for academic research and practical applications. In the future, we intend to explore integrating multimodal data, including visual and auditory information, to enrich the SA process further and address the evolving complexity of online communication. This approach aims to provide a more holistic understanding of sentiments expressed across diverse social media platforms, enhancing the model's applicability and performance in real-world scenarios.

### Funding
The authors received no funding for this work.

### Competing Interests
The authors declare there are no competing interests.

## Author Contributions

- Sonia Khan Papia conceived and designed the experiments, performed the experiments, analyzed the data, performed the computation work, prepared figures and/or tables, authored or reviewed drafts of the article, and approved the final draft.
- Md Asif Khan conceived and designed the experiments, analyzed the data, prepared figures and/or tables, authored or reviewed drafts of the article, and approved the final draft.
- Tanvir Habib conceived and designed the experiments, analyzed the data, prepared figures and/or tables, authored or reviewed drafts of the article, and approved the final draft.
- Mizanur Rahman conceived and designed the experiments, performed the experiments, performed the computation work, authored or reviewed drafts of the article, and approved the final draft.
- Md Nahidul Islam conceived and designed the experiments, performed the experiments, performed the computation work, authored or reviewed drafts of the article, and approved the final draft.

## Data Availability

The IMDB dataset is available at Kaggle: https://www.kaggle.com/datasets/lakshmi25npathi/imdb-dataset-of-50k-movie-reviews. More dataset information is available at: http://ai.stanford.edu/~amaas/data/sentiment.

The Twitter USAirline Data is available at Kaggle: https://www.kaggle.com/datasets/crowdflower/twitter-airline-sentiment.

The code for this experimental analysis is available at Zenodo: Md Nahidul Islam. (2024). nahidul76edu/Sentiment_Analysis-distilrobilstmfuse: Sentiment Analysis-distilrobilstmfuse (v1.0.0). Zenodo. https://doi.org/10.5281/zenodo.13255008.

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
