# Peer review of "DistilRoBiLSTMFuse: an efficient hybrid deep learning approach for sentiment analysis"

_PeerJ Computer Science, doi:10.7717/peerj-cs.2349_

## Round 0.1 · original submission · Major Revisions

Thank you for submitting your manuscript to PeerJ. After a thorough review process, we have received detailed evaluations from our reviewers, and we request that you address the following major revisions before your manuscript can be considered further for publication.

The reviewers have noted that the experimental setup requires a more comprehensive description. Please include specific details about the overall design to enhance the reproducibility and transparency of your research.

It has been suggested that the presentation of your experimental results could be improved with more visual aids such as graphs and tables to clearly illustrate data trends and comparisons. Furthermore, please ensure all key experiments are replicable and statistically analyzed for consistency.

In terms of methodology, the reviewers have emphasized the need for a detailed explanation of the methods used for data analysis, with all statistical tests appropriately referenced and justified.

In addition to addressing these points, please ensure that your revised manuscript includes a point-by-point response letter detailing how each of the reviewers' comments has been addressed, revised figures and tables (if applicable) with improved clarity and accuracy, and an updated discussion section reflecting any changes made in the methodology or experimental evaluation.

We believe that these revisions will significantly strengthen your manuscript and improve its contribution to the field. Please submit your revised manuscript within [timeline, e.g., 3 months] from the date of this email. If you have any questions or require further clarification on any of the points raised, please do not hesitate to contact us.

Reviewer 1 ·

Basic reporting

All comments have been added in detail to the last section.

Experimental design

All comments have been added in detail to the last section.

Validity of the findings

All comments have been added in detail to the last section.

Additional comments

Review Report for PeerJ Computer Science
(DistilRoBiLSTMFuse: an efficient hybrid deep learning approach for sentiment analysis)

1. Within the scope of the study, a deep learning-based model was developed for sentiment analysis using two different open source benchmark datasets.

2. In the introduction section, the importance of the subject and the main contributions of the study, the contributions of sections such as data preprocessing and hybrid architecture are clearly mentioned.

3. In the related works section, studies in the literature on sentiment analysis are examined in terms of deep learning and machine learning models. It is recommended that the literature review here be detailed by adding columns such as "data preprocessing, results, pros and cons" to the existing table.

4. It is positive that more than one (two different) datasets are used instead of relying on a single dataset for the implementation of the proposed model. Dataset selections and the use of data preprocessing operations such as stopwords and data cleaning are sufficient for the study. In addition, the data augmentation section also seems appropriate. However, in the features extraction section, it is stated that The Bag of Word (BoW) model and Term Frequency-Inverse Document Frequency (TF-IDF) are used. Although there are different models that can be used for this in the literature, it should be explained more clearly why these two are preferred.

5. It is stated that a total of five different machine learning models, including Naive Bayes, AdaBoost, and Support Vector Machine, were used as classification models. Although there are many different machine learning models that can be used for classification in the literature regarding the problem solution, it is necessary to state more clearly why these are specifically preferred and/or whether experiments were conducted with different models.

6. When the proposed Hybrid Architecture is examined, it is observed that it has a level of originality and consists of a deep learning-based structure. Experiments on different datasets also reveal the quality of the model. However, it is recommended that the parts such as the framework, toolbox, and hardware used in the application part of the study be explained in more detail.

7. Although metrics such as precision, recall, etc. were obtained in the evaluation metrics section, all other missing metrics such as Kappa statistic, ROC curve and AUC should be obtained for the correct analysis of the results.

As a result, although an important model has been presented in the field of sentiment analysis in the study, the necessary improvements should be made by answering the parts mentioned above in detail and step by step.

·

Basic reporting

no comment

Experimental design

The authors can provide a link to their code (maybe on GitHub) so that the results can be replicated and verified more easily.

Validity of the findings

no comment

Additional comments

1) The authors can consider adding some discussion on whether the approach mentioned will scale easily to other datasets beyond the two mentioned in the paper. For example, can the data prepossessing strategy be readily applied to other datasets or is custom for these two datasets?

2) A number of pages describe details of existing and known concepts in in detail. For example: TF-IDF, Naive Bayes, etc. These sections could be shortened and external references could instead be provided.

3) The authors can consider providing a section on Future Work with some details on how their work can be extended further, or mention any experiments that they would explore in the future.

4) Can the DistilRoBiLSTMFuse model mentioned by called as a part of a single module, and be published/shared as a module/package for future use?

·

Basic reporting

1- In line 151, “… vast amounts of opinionated content generates….”, that expression needs to be reformulated in order to clarify the text.

2- Although in the Results section it is mentioned that the authors split the dataset into 70% training, 15% validation and 15% testing, it is quite important that this explanation appears earlier, specifically in the Methodology section.

3- In line 334, it is mentioned that the hash symbol is removed from the data but in lines 327-330 it is shown that the hash symbol is removed. Therefore, lines 334-337 must be enclosed inside 327-330 to avoid duplication of preprocessing explanations.

4- In line 683 there is a mistake corresponding to “94.16% in testing accuracy” while in Table 5 it is shown that the correct percentage is 93.97%.

5- In line 748, “Table 8 shows the negative class shows exceptionally..”, reformulate that expression to enhance the clarity of the manuscript.

6- In line 760, “After balancing,” the sentence is not complete. It must be completed.

7- In line 807, the name of the model is Bi-LSTM, not BILSTM.

8- In line 822, the Acknowledgements section is empty. If it appears in the paper, it must be filled. However, the authors should delete the name of the section if it remains empty.

Experimental design

1- During the process of splitting the text into training, validation and testing subsets, did the authors use a stratification process to ensure the same proportions of the classes in each training, validation, and testing subset respectively?

2- In the USAirline dataset utilization, were the neutral subsamples from the dataset introduced into the positive group? Were the neutral subsamples not taken into account for the experiments or what actions did the authors take towards these neutral subsamples?

Validity of the findings

1- To give more visual information that helps to understand the results obtained from the model presented, the use of confusion matrices to show the TP, FP, TN and FN obtained from each dataset using the model presented is recommended. Add the confusion matrices as a figure.

2- The authors should make public the repository where the code of these experiments is available. This will contribute to enhancing the scientific contribution of this research and can help other researchers to replicate the experiments.

---

## Round 0.2 · Minor Revisions

Thank you for submitting your manuscript to PeerJ Computer Science. The re-review process has now been completed, and we have received the feedback from the reviewers.

I am pleased to inform you that, while the reviewers found your work to be of significant interest and quality, they have also identified a limited number of revisions that could further enhance the overall quality and clarity of the paper. These suggested revisions are minor but important for ensuring that your manuscript meets the highest standards of the journal.

We therefore invite you to submit a revised version of your manuscript that addresses the reviewers' comments. Please include a detailed response to each point raised by the reviewers, explaining how you have addressed their feedback or providing a justification if you have not made certain changes.

Once we receive your revised manuscript, we will proceed with the final stages of the review process.

Reviewer 1 ·

Basic reporting

All comments have been added in detail to the last section.

Experimental design

All comments have been added in detail to the last section.

Validity of the findings

All comments have been added in detail to the last section.

Additional comments

Review Report for PeerJ Computer Science
(DistilRoBiLSTMFuse: an efficient hybrid deep learning approach for sentiment analysis)

Thank you for the revision. The relevant changes made to the paper after the first evaluation and the responses to the referee comments are appropriate. I recommend that the paper be accepted both because of its potential to make a significant contribution to the literature and its originality. I wish the authors success in their future studies. Kind regards.

·

Basic reporting

The authors have successfully fulfilled most of the considerations that were provided. However, there is one aspect that must be addressed:

1. When the authors refer to the model called Bi-LSTM, they must ensure that it is written correctly. Please review the entire paper to ensure that whenever this model is mentioned, it is written correctly."

Experimental design

No comment

Validity of the findings

No comment

Additional comments

No comment

---

## Round 0.3 · accepted · Accept

I am pleased to inform you that after careful consideration, your manuscript has been accepted for publication in PeerJ Computer Science. The reviewers have thoroughly evaluated your submission and have provided constructive feedback, which has been successfully addressed in the revised version of your manuscript.

The revisions you have made have enhanced the clarity and quality of the paper, and we are satisfied that the work is now of a high standard, ready for publication. The thoroughness of the review process and your diligent responses to the reviewers' comments have ensured that the final version of your manuscript meets the rigorous standards of our journal.

Thank you for choosing PeerJ Computer Science as the venue for your research.

·

Basic reporting

no comment

Experimental design

no comment

Validity of the findings

no comment

Additional comments

The authors have successfully addressed all the required aspects of the revisions.